# SPLIT AND MERGE PROXY: PRE-TRAINING PROTEIN INTER-CHAIN CONTACT PREDICTION BY MINING RICH INFORMATION FROM MONOMER DATA

## ABSTRACT

Protein inter-chain contact prediction is a key intelligent biology computation technology for protein multimer function analysis but still suffers from low accuracy. An important problem is that the number of training data cannot meet the requirements of deep-learning-based methods due to the expensive cost of capturing structure information of multimer data. In this paper, we solve this data volume bottleneck in a cheap way, borrowing rich information from monomer data. To utilize monomer (single chain) data in this multimer (multiple chains) problem, we propose a simple but effective pre-training method called Split and Merge Proxy (SMP), which utilizes monomer data to construct a proxy task for model pre-training. This proxy task cuts monomer data into two sub-parts, called pseudo multimer, and pre-trains the model to merge them back together by predicting their pseudo contacts. The pre-trained model is then used to initialize our target – protein inter-chain contact prediction. Because of the consistency between this proxy task and the final target, the whole method brings a stronger pre-trained model for subsequent fine-tuning, leading to significant performance gains. Extensive experiments validate the effectiveness of our method and show the model performs better than the state-of-the-art (SOTA) method by 11.40% and 2.97% on the P@$L/10$ metric for bounded benchmarks DIPS-Plus and CASP-CAPRI, respectively. Further, the model also achieves almost 1.5 times performance superiority to the SOTA approach on the harder unbounded benchmark DB5. Finally, we also effectively apply our SMP on docking and interaction site prediction tasks to verify the SMP is a general method for other multimer-related tasks. The code, model, and pre-training data will be released after this paper is accepted.

## 1 INTRODUCTION

Proteins are large molecules consisting of amino acid (also called residue) sequences. Protein inter-chain contact prediction aims to compute the binding between chains for given protein sequences (specifically whether an individual amino acid on one chain is in contact with residue on the other chain), which is important for the structural or functional analysis of protein complexes. The predicted binding reveals the geometric relationships between each residue pair of the two chains, which can not only benefit multimer structure prediction but also be useful for many kinds of protein function analysis scenarios, e.g. developing new drugs and designing new proteins. The success of RaptorX (Wang et al., 2017; Xu et al., 2021) and AlphaFold2 (Jumper et al., 2021) demonstrates the application potential of deep learning in the computational biology field and inspired a series of new biological computation methods. However, when extending the deep model to protein inter-chain contact prediction, recent works have not achieved satisfying performance as the aforementioned successful works do. An important bottleneck is data quantity limitation.

Many well-known successful deep learning systems are trained under large-scale datasets. For example, in computer vision (CV), ConvNet (Krizhevsky et al., 2012; Simonyan & Zisserman, 2014; He et al., 2016)) and ViT (Dosovitskiy et al., 2020; Liu et al., 2021; Yuan et al., 2021) are trained on ImageNet (Deng et al., 2009) which has 14 million labeled data who provide enough vision category information of real word. For natural language processing (NLP), the most popular language model BERT (Devlin et al., 2018) is trained on document-level data BooksCorpus (Zhu et al., 2015) and English Wikipedia with 3,300 million words in an unsupervised manner. In computational biology, the

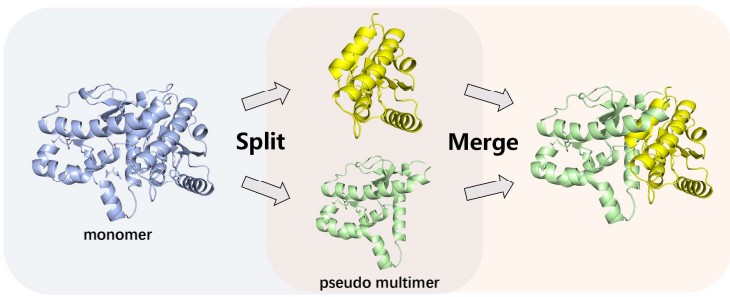

Figure 1: The main idea of Split and Merge Proxy (best viewed in color). In the pre-training stage, a monomer (single chain) is firstly split into two sub-parts that are treated as pseudo multimers (a pair of chains). Then the deep model is trained by learning to merge the pseudo multimers back through predicting their inter-chain contacts.

recent most famous protein structure prediction model AlphaFold2 (Jumper et al., 2021) is trained on about 400k monomer data, 60k with 3D structure labels of Protein Data Bank (PDB) (wwp, 2019) and 350k protein sequence, and achieves electron microscope accuracy. Obviously, existing human-level accurate and successful artificial intelligence models also need big data to train. However, the number of the current largest open-sourced multimer training data is much lower than the aforementioned topics. For instance, there is only 15k training data in the protein inter-chain contact prediction task which could limit the performance of the deep model. The main reason is the expensive cost of capturing the protein complex structural information by high-accurate devices. So to tackle the problem of the scarcity of training data, we focus on finding a cheap way to obtain additional data and avoid the extra cost.

Our main idea is to expand the training data by introducing the monomer data into the training step for protein inter-chain contact prediction. The existing monomer data is free and also can provide useful biological prior. Some works (Zeng et al., 2018; Bryant et al., 2022; Gao et al., 2022) introduce the monomer data into the multimer-related tasks. These works treat the multimer as a monomer and try to directly feed it into a monomer-based model without any modification. It proves the potential value of the monomer data to the multimer task. But obviously, there is an unneglectable task gap between the monomer and the multimer. Specifically, the monomer can only provide information about one chain while the multimer task requires more. So the above methods suffer from that task gap and existing contact prediction methods often neglect these data. In this paper, we design a novel and effective pre-training method called Split and Merge Proxy (SMP) to introduce monomer data into the protein inter-chain contact prediction task more effectively, which reduces the aforementioned task gap and leads to better results.

The proposed SMP is a proxy task for contact prediction pre-training. As shown in Figure 1, SMP generates pseudo multimer data from monomers and utilizes that data to pre-train the contact prediction model. In particular, a protein with a single chain is **split** into two sub-parts that are treated as a pseudo multimer. That pseudo data are used to train the contact prediction model, equal to guide the model to **merge** these split data back. Although the pseudo multimer data contain biological noise, they can provide additional richer information that complements the existing multimer data. The training targets of SMP and the final task are both contact predictions, so there is no task gap in the fine-tuning stage. The pre-trained model can be fine-tuned on the real multimer data without any modification, leading to a better final model and more accurate contact results.

Our main contributions are as follows:

- We design a novel proxy task, Split and Merge Proxy (SMP), to pre-train contact prediction models on the monomer data more effectively. To the best of our knowledge, this is the first work to leverage the monomer data to pre-train the multimer contact prediction task.

- Experiments show that we achieve a new state of the art and improve the P@ $L/10$ metric by a large margin – 11.40% and 2.97% respectively on DIPS-Plus and CASP-CAPRI benchmarks when compared with the SOTA GeoTrans (Morehead et al., 2022). Moreover, we almost achieve 1.5 times more performance than GeoTrans on the harder unbounded benchmark DB5.

- We also use the SMP for other multimer-related tasks, especially protein docking and protein interaction site prediction. The experiments indicate that the SMP could effectively

improve the results of the current popular model, which demonstrates that the SMP is a general method that can be combined with other methods to use in multimer-related tasks.

## 2 RELATED WORKS

**Protein inter-chain contact prediction.** Protein intra-chain contact prediction has been well treated in the past methods (Jumper et al., 2021; Baek et al., 2021), but protein inter-chain contact prediction has not been extensively studied. Some early works (Weigt et al., 2009; Morcos et al., 2011; Ekeberg et al., 2014) used direct-coupling analysis (DCA) to disentangle direct and indirect correlations to infer potential relationships between amino acids at different positions. With the great success of Convolutional Neural Network (CNN) (LeCun et al., 1998) in CV area, Zeng et al. (2018); Yan & Huang (2021); Roy et al. (2022); Lin et al. (2023) applied CNN to multimer contact prediction. Zeng et al. (2018) used two CNNs, one with 1D convolution processed sequence information and the other with 2D convolution encoded multimer multiple sequence alignment (MSA) information. Yan & Huang (2021); Lin et al. (2023) utilized more biological features (e.g., inter-protein docking pattern, physico-chemical information, and sequence conservation) as inputs to the neural network to enrich the information carried by multimer data. Because He et al. (2016) demonstrated that deeper networks could learn more discriminative features from the dataset, Roy et al. (2022); Guo et al. (2022b) used a deeper dilated residual network (Yu et al., 2017) to capture relationships between residues. Due to each protein has a 3D structure, Fout et al. (2017); Liu et al. (2020); Morehead et al. (2022); Xie & Xu (2022) designed graph neural network (GNN) (Scarselli et al., 2008) to predict contacts between chains. They first built a graph for each protein, the residue on each protein is regarded as a node, and whether the residues in the protein are connected is regarded as an edge. Fout et al. (2017) used graph convolution (Kipf & Welling, 2016) to get the graph representation of the underlying protein structure and a fully convolutional network (FCN) was utilized to determine contacts between two proteins. Liu et al. (2020) employed weights-sharing GNNs to obtain the residue features of each protein, then they devised multilayer CNNs as the interaction module to perform contact prediction. Based on this, Morehead et al. (2022) designed graph transformers to encode the geometric information in multimers, e.g., the distance and direction between residues and the amide angle. Xie & Xu (2022) believed that simply building the residue graph was not enough, so they built two more graphs, e.g., atom graph and surface graph, then they did message passing in each graph. Since AlphaFold2 (Jumper et al., 2021) has achieved surprising results in monomer structure prediction, Evans et al. (2021); Bryant et al. (2022); Gao et al. (2022) extended it to multimer contact prediction. Evans et al. (2021) took into account permutation symmetry, position encoding of different chains in multimer, and multimer MSA construction for contact prediction. Bryant et al. (2022); Gao et al. (2022) directly spliced multimer as monomer and fed it into AlphaFold2 to get contact prediction. However, due to the small scale of existing multimer data, current models are less accurate in protein inter-chain contact prediction.

**Pre-training in protein modeling.** Pre-training from a lot of data can provide good prior knowledge for the model, so it achieves great success in the data science community, such as CV and NLP areas. Some recent works introduced the pre-training paradigm to the protein modeling area. Rao et al. (2021); Rives et al. (2021); Elnaggar et al. (2021); Chowdhury et al. (2021); Fang et al. (2022); Lin et al. (2022) used Masked Language Model (MLM) proxy task (Devlin et al., 2018) to learn residue embedding from massive protein sequences. Rives et al. (2021); Elnaggar et al. (2021); Chowdhury et al. (2021) directly utilized transformer (Vaswani et al., 2017) as a pre-training network to capture potential biological patterns of amino acids. Since MSAs can provide a certain biological prior for the model, Fang et al. (2022); Lin et al. (2022) devised the same Evoformer network as AlphaFold2 (Jumper et al., 2021) and Rao et al. (2021) designed MSA transformer to fully integrate the MSA information into the transformer architecture in pre-training stage, which can make the network directly learn evolutionary information. Because each atom of the protein in PDB Database (wwp, 2019) has 3D coordinates, Gligorijević et al. (2021); Chen et al. (2022) designed distance prediction and the dihedral angle prediction proxy tasks, then they got the underlying structural representations for monomers and achieved excellent performance in protein classification tasks. Guo et al. (2022a) thought the natural protein coordinates are noisy, they proposed a proxy task that estimates the gradient of the perturbed 3D structure of the protein. The model used SE(3)-invariant representation as the inputs and got better results on protein structure quality assessment and protein interaction site prediction tasks. Zhang et al. (2022) used the multiview contrastive learning and self-prediction tasks to pre-train the monomer graph encoder, they outperformed the past methods in function prediction and fold classification tasks. Due to the lack of multimer data and the cost of collecting multimer data is expensive, it is difficult to build an effective pre-training paradigm on

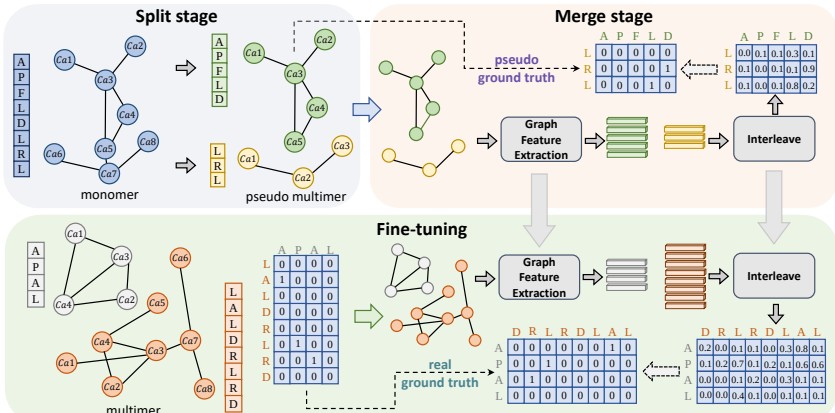

Figure 2: The Framework of the proposed Split and Merge Proxy (SMP) pre-training method (best viewed in color). The split stage cuts a monomer (the sequence "APFLDLRL" in the figure) into a pseudo multimer consisting of two sub-parts (the two sub-parts "APFLD" and "LRL" in the figure) and computes the contact ground truth. The merge step is the pre-training process, which trains the model to predict the contact relationships on the split data, essentially equal to merging the split sequences back. Note that the sequence here is just an example, not the real chain. In the fine-tuning stage, the whole contact predictor, including the graph feature extractor and the interleave module, is directly fine-tuned without any modification on the real multimer data.

existing multimer data. So in this paper, we design a novel proxy task to adapt the monomer into multimer contact prediction, which can pre-train the model to get stronger performance.

## 3 METHOD

### 3.1 TASK DEFINITION

The protein inter-chain contact prediction aims to compute the contact relationship map $A \in (0,1)^{L_1 \times L_2}$ between the two given protein chains. The element $A_{i,j}$ is 0 or 1, indicating whether the $i$-th residue in one sequence interacts with the $j$-th residue in another sequence or not. The contact prediction models take multiple kinds of biological features as inputs, such as amino acid sequences $s \in \mathbb{P}^L$ ($\mathbb{P}$ is the set of the amino acid, including 20 kinds of amino acids) and residue 3D coordinates $c \in \mathbb{R}^{L \times 3}$ which is essentially the location of the non-hydrogen or carbon-alpha (C$\alpha$) atoms. The computation pipeline can be defined as:

$$A^{pred} = f(x_1, x_2), \quad \text{where} \quad x_i = \{s_i, c_i\}, \quad i \in \{1, 2\} \tag{1}$$

To achieve this, whatever details of the function $f$, proteins are often regarded as a graph so the residues are treated as nodes and processed by graphic models to extract features. After that, an interleave module fuses these node features and measures the similarities between each residue pair to calculate the contact scores. The whole process is the same as the Fine-tuning block in Figure 2.

### 3.2 SPLIT AND MERGE PROXY

The Split and Merge Proxy (SMP) is an effective proxy task proposed to pre-train the contact prediction model. The main pipeline includes a split step and a merge step, shown in Figure 2. Each monomer sequence is cut into two sub-parts to generate the pseudo multimer data. In the merge step, the model learns the contact prediction task directly on the aforementioned split data without any modification. After that, the model would be fine-tuned on real multimer data.

**Split stage:** We use the monomers from the PDB (wwp, 2019) dataset because of its monomer data including both amino acid sequences and corresponding 3D coordinates. The target of the split stage is to generate the pseudo multimer that has the same data structure as the real one, including two sub-sequences ($s_1$ and $s_2$) with structural information ($c_1$ and $c_2$) and their corresponding contact ground truth $A$. We first cut the amino acid sequence into two sub-sequences at a random location:

$$s_1 = s[:l], s_2 = s[l:], \tag{2}$$

where $l$ means the random split index uniformly sampled from the range $R$, keeping each cut sequence informative and avoiding too short split results that contain only small amounts of residues. In other words, this split location is around the center of the given sequence.

And for the 3D structure, we do a similar split operation:

$$c_1 = c[: l], c_2 = c[l :]. \tag{3}$$

We do not operate any normalization on these 3D coordinates, keeping their values still in the monomer coordinate system so ground-truth $A$ could be computed by the following formula:

$$A_{i,j} = \begin{cases} 1 & D_{i,j} \le \lambda \\ 0 & D_{i,j} > \lambda \end{cases} \quad , \quad D_{i,j} = ||c_1[i] - c_2[j]||^2, \tag{4}$$

where $\lambda$ is the threshold to determine whether the $i$-th and $j$-th residue pair contact or not. $||\cdot||$ means the Euclidean distance. This process could be interpreted that the ground-truth contact of pseudo multimer is equal to the intra-chain contact of the original monomer. Based on the steps mentioned above, monomer data is converted to the pseudo multimer in the form of $\{s_1, s_2, c_1, c_2, A\}$.

**Merge stage:** The merge stage is essentially the mimicking learning of the standard contact prediction training. The model learns to predict the $A$ based on the given pseudo multimer inputs $\{s_1, s_2, c_1, c_2\}$. Specifically, We first extract the sequence and structure information for each cut sub-sequence by MSA and Protein Structure and Interaction Analyzer (PSAIA), respectively. And then, these pieces of information combined with the original protein sequence and 3D structural information are sent to a graph feature extractor to extract residue features $F_1 \in \mathbb{R}^{L_1 \times C}$ and $F_2 \in \mathbb{R}^{L_2 \times C}$ like Figure 2 shows. Note that the coordinate values in $c_1, c_2$ all belong in the same monomer coordinate system. So they are all treated to the relative distances of residue pairs in each protein sequence to avoid information leakage. After that, an interleave module computes the interaction features $F_I \in \mathbb{R}^{L_1 \times L_2 \times C'}$, which stores the high-level relationship patterns for each residue pair. Finally, a contact prediction head, often a fully convolutional neural network (FCN), predicts a contact map based on those features. For the prediction, we train it as a binary classification task by utilizing the cross-entropy loss function.

**Fine-tuning stage:** The SMP task is the same as the final contact prediction task, both predicting the protein inter-chain contact maps. So there is not any task gap between this proxy task and fine-tuning. Every module and parameter of the pre-trained model could be re-used in the final model. So, We feed the real multimer data into the pre-trained model and fine-tune the whole model directly.

## 4 EXPERIMENTS

### 4.1 DATASET AND EVALUATION PROTOCOL

In this section, we conduct several experiments on three popular benchmarks DIPS-Plus (Morehead et al., 2021), CASP-CAPRI (Lensink et al., 2019; 2021) and DB5 (Vreven et al., 2015) datasets.

**DIPS-Plus** is the latest open-sourced dataset for protein inter-chain contact prediction. It provides amino acid sequences and residue coordinates for each multimer data. Except for these pieces of basic information, DIPS-Plus also offers additional different types of biological features such as protrusion index and amide plane normal vector, composing much richer information. After filtering extreme data, such as too long, too short sequences and high relative data with other datasets, the DIPS-Plus dataset still has 15,618 and 3,548 protein complexes for training and validation, respectively, which is the recent known largest open-sourced benchmark. For testing, it provides 32 protein complexes consisting of 16 homodimers and 16 heterodimers to evaluate the model's ability to handle samples of different difficulties.

**CASP-CAPRI** has been well known as a biologically joint challenge since 2014, aiming to assess the computational methods of modeling protein structures. Morehead et al. (2022) re-organized the data of the 13th and 14th CASP-CAPRI challenge sessions (Lensink et al., 2019; 2021), filtering the overlap between the original CASP-CAPRI data and the DIPS-Plus. These filtered data include 14 homodimers and 5 heterodimers and are used to evaluate the ability of real-world applications and cross-set generalization of models trained on the DIPS-Plus training set.

**DB5** (Docking Benchmarks version 5 (Vreven et al., 2015)) is a traditional benchmark for inter-chain contact prediction, including 140 training, 35 evaluation, and 55 testing samples. DB5 consists of unbounded protein complexes that have varying contact types. In contrast, complexes in DIPS-Plus and CASP-CAPRI are bounded and their multiple chains are already conformed with each other. So it can indicate the performance and effectiveness of our model on different types of complexes.

Table 1: The average top-k precision (P@k) and recall (R@k) on DIPS-Plus test dataset (%).

| | 16 (Homo) | | | 16 (Hetero) | | |
|---|---|---|---|---|---|---|
| Method | P@ $L/10$ | P@ $L/5$ | P@ $L/2$ | P@ $L/10$ | P@ $L/5$ | P@ $L/2$ |
| BIPSPI (Sanchez-Garcia et al., 2018) | 0 | 0 | - | 2.00 | 2.00 | - |
| DeepHomo (Yan & Huang, 2021) | 12.00 | 9.00 | - | - | - | - |
| ComplexContact (Zeng et al., 2018) | - | - | - | 16.00 | 15.00 | - |
| GCN (Morehead et al., 2022) | 20.00 | 18.00 | - | 8.00 | 7.00 | - |
| GeoTrans (Morehead et al., 2022) | 25.00 | 23.00 | - | 14.00 | 11.00 | - |
| GeoTrans + SMP | **39.81** | **33.33** | **26.02** | **20.99** | **20.07** | **15.00** |
| | 32 (All Proteins) | | | | | |
| Method | P@ $L/10$ | P@ $L/5$ | P@ $L/2$ | R@ $L$ | R@ $L/2$ | R@ $L/5$ |
| BIPSPI (Sanchez-Garcia et al., 2018) | 1.00 | 1.00 | - | 1.00 | 0.40 | 0.30 |
| GCN (Morehead et al., 2022) | 16.00 | 12.00 | - | 10.00 | 6.00 | 3.00 |
| GeoTrans (Morehead et al., 2022) | 19.00 | 17.00 | - | 15.00 | 9.00 | 4.00 |
| GeoTrans + SMP | **30.40** | **26.70** | **20.51** | **24.00** | **16.02** | **8.56** |

Table 2: The average top-k precision and recall on CASP-CAPRI 13 & 14 dataset.

| | 14 (Homo) | | | 5 (Hetero) | | |
|---|---|---|---|---|---|---|
| Method | P@ $L/10$ | P@ $L/5$ | P@ $L/2$ | P@ $L/10$ | P@ $L/5$ | P@ $L/2$ |
| BIPSPI (Sanchez-Garcia et al., 2018) | 0 | 0 | - | 0 | 3.00 | - |
| DeepHomo (Yan & Huang, 2021) | 2.00 | 2.00 | - | - | - | - |
| ComplexContact (Zeng et al., 2018) | - | - | - | 8.00 | 5.00 | - |
| GCN (Morehead et al., 2022) | 11.00 | 13.00 | - | 11.00 | 9.00 | - |
| GeoTrans (Morehead et al., 2022) | 13.00 | 11.00 | - | 31.00 | **24.00** | - |
| GeoTrans + SMP | **18.63** | **14.37** | **11.57** | **32.00** | 23.49 | **18.35** |
| | 19 (All Proteins) | | | | | |
| Method | P@ $L/10$ | P@ $L/5$ | P@ $L/2$ | R@ $L$ | R@ $L/2$ | R@ $L/5$ |
| BIPSPI (Sanchez-Garcia et al., 2018) | 0 | 1.00 | - | 2.00 | 1.00 | 0.1 |
| GCN (Morehead et al., 2022) | 10.00 | 9.00 | - | 11.00 | 6.00 | 2.00 |
| GeoTrans (Morehead et al., 2022) | 19.00 | 14.00 | - | 13.00 | 8.00 | **4.00** |
| GeoTrans + SMP | **21.97** | **16.77** | **13.36** | **14.33** | **8.34** | 3.91 |

**Evaluation** All the experiments follow the standard evaluation protocol in existing inter-chain contact prediction benchmarks. To assess the accuracy of the prediction, the top-$k$ precision and recall are adopted as the evaluation metrics, where $k \in \{L/30, L/20, L/10, L/5, L/2, L\}$ with $L$ is the length of the shortest chain.

## 4.2 IMPLEMENTATION DETAILS

We generate the pseudo multimer from all monomers before 2018-4-30 from PDB (wwp, 2019). There are 60,206 data in total. Each file contains sequence and structural information for the protein. Monomers that cannot be parsed by Biopython (Cock et al., 2009) (containing unknown atoms; missing atoms; chain numbers are not in order and so on) are filtered out. Except that each protein file contains several conformations, we only keep the first one and abandon the other. We set the split range $R = \{1/3 \sim 2/3\}$ so that the cut position is close to the middle of the given sequence to get pseudo multimers. Too short split proteins whose length of any chain is less than 20 are dropped. The threshold $\lambda$ used to calculate the contact ground truth is set as 6 Å following the same procedure that real multimer utilizes in Morehead et al. (2021). Finally, there are 22,589 pseudo multimers, about 1.5 times the existing real multimer contact dataset. Due to the ID numbers of monomer and multimer in PDB being different, there is no overlap between pseudo multimer and real multimer data. We also discuss the potential leakage of pre-training data in the appendix. Whatever for the pseudo or real multimer data, we all use HHBlits (Remmert et al., 2012) with Uniclust30 (Mirdita et al., 2017) database for MSA, and PSAIA (Mihel et al., 2008) to calculate geometric features.

Our SMP is a pre-training method that is not tightly bound to a specific model. So we combine SMP with the GeoTrans (Morehead et al., 2022) to evaluate the effectiveness of SMP in the following experiments. The batch size of pre-training and fine-tuning are all set as 48 (except the fine-tuning one of CASP-CAPRI is set as 32 because of the cross-domain evaluation setting of CASP-CAPRI). Other experimental settings, including loss function, learning rate, and so on, are all kept the same as the latest open-sourced SOTA GeoTrans. More implementation details can be seen in the appendix.

## 4.3 COMPARISON WITH SOTA METHODS

We compare several SOTA multimer contact prediction methods including BIPSPI (Sanchez-Garcia et al., 2018), ComplexContact (Zeng et al., 2018), DeepHomo (Yan & Huang, 2021), GCN (Morehead et al., 2022) and GeoTrans (Morehead et al., 2022). Except that the input of ComplexContact is the amino acid sequence, the other methods take both amino acid sequence and 3D structural information as inputs, which are the same as our model.

Table 1 shows the comparison results between SMP and other methods on the DIPS-Plus dataset, demonstrating that SMP outperforms existing SOTA GeoTrans by a large margin. For homologous complexes, SMP outperforms GeoTrans by 10.33% on the harder metric P@ $L/5$ and even 14.81% on P@ $L/10$, demonstrating that SMP can learn more useful residue representation and contact

Table 3: The average top-k precision and recall on DB5 test dataset.

| | 55 (Hetero) | | | | | |
|---|---|---|---|---|---|---|
| Method | P@ $L/10$ | P@ $L/5$ | P@ $L/2$ | R@ $L$ | R@ $L/2$ | R@ $L/5$ |
| BIPSPI (Sanchez-Garcia et al., 2018) | 0.20 | 0.10 | - | 0.30 | 0.10 | 0.04 |
| ComplexContact (Zeng et al., 2018) | 0.30 | 0.30 | - | 0.70 | 0.30 | 0.10 |
| GCN (Morehead et al., 2022) | 0.60 | 0.70 | - | 1.30 | 0.80 | 0.30 |
| GeoTrans (Morehead et al., 2022) | 0.90 | 1.10 | - | 1.80 | 1.00 | 0.34 |
| GeoTrans + SMP | **1.78** | **1.88** | **1.55** | **2.53** | **1.45** | **0.69** |

Table 4: SMP vs self-supervised pre-training (SSL) on DIPS-Plus test dataset.

| Row | Model | PreTrain | P@ $L/10$ | P@ $L/5$ | P@ $L/2$ | P@ $L$ | R@ $L$ | R@ $L/2$ | R@ $L/5$ | R@ $L/10$ |
|---|---|---|---|---|---|---|---|---|---|---|
| 1 | GCN | - | 16.00 | 12.00 | - | - | 10.00 | 6.00 | 3.00 | - |
| 2 | GCN | SMP | 18.96 | 15.64 | 11.61 | 8.24 | 13.58 | 10.04 | 5.36 | 3.14 |
| 3 | GeoTrans | - | 19.00 | 17.00 | - | - | 15.00 | 9.00 | 4.00 | - |
| 4 | GeoTrans | Mask-node (Hu et al., 2020) | 20.87 | 18.19 | 14.62 | 12.40 | 17.46 | 9.88 | 4.87 | 2.83 |
| 5 | GeoTrans | Mask-edge (Hu et al., 2020) | 20.26 | 17.67 | 14.31 | 11.34 | 16.37 | 10.47 | 5.16 | 2.71 |
| 6 | GeoTrans | PHD (Li et al., 2021) | 20.19 | 17.32 | 14.50 | 11.05 | 16.51 | 10.75 | 5.34 | 2.83 |
| 7 | GeoTrans | SMP | **30.40** | **26.70** | **20.51** | **15.87** | **24.00** | **16.02** | **8.56** | **4.79** |

prediction knowledge from additional pseudo multimer data. For more difficult heterologous complexes, SMP also surpasses GeoTrans 9.07% on harder P@ $L/5$. These heterologous performances benefit from the potential consistency with the pseudo multimer and heterologous proteins. Specifically, the cut chains usually have low sequence identities, sharing certain similar properties and distributions of the real heterologous data, making SMP an obvious improvement on heterologous multimers. From an overall perspective, the proposed SMP brings significant gains compared with GeoTrans by 11.40% at P@ $L/10$ and 9.00% at R @$L$, proving that SMP brings more discriminative expression for multimer contact prediction whatever homologous or heterologous complexes.

Table 2 presents the average top-$k$ metrics of SMP on the CASP-CAPRI dataset, specifically, 19 challenging protein complexes (14 homodimers and 5 heterodimers). SMP also surpasses the state-of-the-art method GeoTrans on P@ $L/10$ by 5.63% on 14 homologous when keeping comparable performances for 5 heterologous. SMP achieves improvements for several different settings, demonstrating that the pre-training of SMP learns many valuable patterns of contact prediction from pseudo multimers to help learn real multimer prediction effectively.

On the DB5 dataset in Table 3, SMP also exceeds the precision of GeoTrans for all metrics. All methods perform poorly due to testing hard and unseen unbound complexes with varying contact types that are not necessarily conformal. However, SMP still shows more than 1.5 times better performance than GeoTrans in almost all metrics. It indicates that SMP has good cross-domain capabilities and has the potential to be used in real-world applications of complex contact prediction.

Overall, this pre-training paradigm plays a considerable role in various types of downstream multimer contact prediction tasks (cross set and unbound set), showing good robustness with SMP. We also compare the SMP with the AlphaFold series model, the results can be seen in the appendix.

## 4.4 ABLATION STUDIES

### 4.4.1 COMPARISON WITH DIFFERENT PRE-TRAINING PARADIGM AND CONTACT PREDICTOR

Previous comparisons show the effectiveness of the combination of our SMP with the SOTA Geo-Trans. In this ablation study, we further investigate the superiority of our SMP. We combine SMP with different contact predictors to prove its generalization and also compare the SMP with other pre-training methods to show the advantage of the SMP design. All results are shown in Table 4.

To investigate the influence of the combined contact predictor with SMP, we change the graph feature transfer module from the Transformer into the GCN (Kipf & Welling, 2016). This GCN only has a total of 33K parameters, which is quite much lower than the 1.4M parameters of the Transformer one. So this setting can show the generalization of the SMP on a small-scale model. From the 1st and 2nd lines of Table 4, it can be seen that our SMP still brings a 3.64% performance increase under the P@ $L/5$. It indicates that the SMP paradigm keeps strong generalization on the small-scale model, showing the potential for extensions of future different types and levels of contact predictors.

To show the superiority of the SMP design, we construct another experiment to pre-train the graph encoder by adapting popular mask modeling paradigm (Hu et al., 2020) and the PHD method (Li et al., 2021) on the monomer data with 3D structural cues. The mask paradigm pre-trains the model by reconstructing the masked parts through partial unmasked observation. The PHD method defines the proxy task as discriminating whether two half-graphs are derived from the same source or not. These approaches provide different pre-training mechanisms compared with our SMP. As shown in

Table 5: Partial pre-training and fine-tuning results on DIPS-Plus test dataset.

| Row | Ratio | Partial pre-training w/ full-finetuning | | | | Partial fine-tuning w/ full-pretraining | | | |
|---|---|---|---|---|---|---|---|---|---|
| | | P@ $L/10$ | P@ $L/5$ | R@ $L/2$ | R@ $L$ | P@ $L/2$ | P@ $L$ | R@ $L/5$ | R@ $L/10$ |
| 1 | 0 | 19.00 | 17.00 | 9.00 | 15.00 | 2.08 | 1.98 | 0.62 | 0.56 |
| 2 | $1/5$ | 18.22 | 15.58 | 10.48 | 16.80 | 12.25 | 9.63 | 4.20 | 1.96 |
| 3 | $1/4$ | 18.61 | 19.02 | 11.14 | 17.08 | 12.53 | 10.80 | 4.91 | 3.10 |
| 4 | $1/3$ | 24.64 | 21.36 | 12.08 | 16.93 | 12.96 | 10.62 | 4.56 | 2.50 |
| 5 | $1/2$ | 26.20 | 21.29 | 11.50 | 18.09 | 15.49 | 11.92 | 5.53 | 3.22 |
| 6 | 1 | **30.40** | **26.70** | **16.02** | **24.00** | **20.51** | **15.87** | **8.56** | **4.79** |

Table 6: Different split ranges results on DIPS-Plus validation dataset.

| Row | Range | P@ $L/10$ | P@ $L/5$ | P@ $L/2$ | P@ $L$ | R@ $L$ | R@ $L/2$ | R@ $L/5$ | R@ $L/10$ |
|---|---|---|---|---|---|---|---|---|---|
| 1 | $2/5 \sim 3/5$ | 46.57 | 43.62 | 36.42 | 28.16 | 41.89 | 28.91 | 14.72 | 7.84 |
| 2 | $1/3 \sim 2/3$ | **49.82** | **46.44** | **38.02** | **28.71** | **43.28** | **30.63** | **15.84** | **8.49** |
| 3 | $1/4 \sim 3/4$ | 49.14 | 45.66 | 37.25 | 28.25 | 42.07 | 29.58 | 15.41 | 8.27 |
| 4 | $1/5 \sim 4/5$ | 41.90 | 39.08 | 32.34 | 24.75 | 37.53 | 25.95 | 13.30 | 7.08 |

the 3rd $\sim$ 6th lines of Table 4, The mask modeling and PHD method provide average 1% gains on all metrics, proving the fact that monomers bring much useful information for this multimer task from a different view. When compared with the SMP (7th line), SMP still shows stronger performance and outperforms the mask paradigm by 5.89% on the harder metric P@ $L/2$, indicating the superiority of the SMP design that can utilize information in 3D structures more effectively and further eliminate the task gap between the pre-training and fine-tuning stage.

### 4.4.2 PARTIAL PRE-TRAINING RESULTS

We study the effectiveness of pre-training data volume for SMP and conduct partial pre-training experiments with different degrees of monomer data. We set five partial pre-training ratios $\{1/5, 1/4, 1/3, 1/2, 1\}$. When comparing the 2nd line of Table 5 with the 1st line (without pre-training), we find that the performance has some fluctuation when the number of introduced pseudo multimers is small. This is caused by the biological noise introduced by the small-scale pseudo data, which is eliminated when the scale increases and clearly indicated in Table 5 3rd$\sim$6th lines. Obviously, when the amount of pre-trained data reaches $1/4$ (in the 3rd line), SMP has introduced certain precision and recall gains except on P@ $L/10$ metric than GeoTrans (in the 1st line), with an average improvement of 2%. Moreover, as the amount of pre-trained data increases, the performance gradually improves, proving that SMP guides the model to learn rich contact prediction to provide beneficial initialization parameters for contact prediction models.

### 4.4.3 PARTIAL FINE-TUNING RESULTS

The pre-trained model has the potential to achieve satisfying performances only trained with small-scale training data. So we aim to explore the effect of SMP for fine-tuning with different scale data. We use six partial fine-tuning ratios, which belong to the set $\{0, 1/5, 1/4, 1/3, 1/2, 1\}$. The 1st line of Table 5 shows that SMP surpasses the traditional method BIPSPI (Table 1) without any fine-tuning, which indicates that pseudo multimer can provide prior knowledge that is relevant to the real multimers contact prediction. Moreover, the 3rd line of Table 5 shows that our model achieves comparable results to the SOTA predictor GeoTrans only with 1/4 training data demonstrating that the SMP pre-training can provide knowledge that can be effectively re-used and transferred to the real multimer scenario. With further increasing the data volume in the 4th $\sim$ 6th line of Table 5, it can finally achieve 8.56% on metric R@ $L/5$, surpassing the previous SOTA GeoTrans. These experiments prove that our pre-training paradigm can effectively reduce the dependence on real data and make the model adapt to different volume-level training data situations, having the potential to save the extra cost of collecting multimer data. We provide more detailed results in the appendix.

### 4.4.4 DIFFERENT SPLIT RANGE RESULTS

We study the influence of different split ranges on the split stage for SMP to find the optimal one. We set four split intervals settings $\{2/5 \sim 3/5, 1/3 \sim 2/3, 1/4 \sim 3/4, 1/5 \sim 4/5\}$ on the validation set. Because the validation set has more samples than the test set (3548 v.s 32), it could provide more stable results. As shown in the 2nd line of Table 6, we find the performance is best when the split interval is $1/3 \sim 2/3$. The other split positions have similar results except for the range $1/5 \sim 4/5$. Because splitting at $1/5 \sim 4/5$ would cause some too-short chains which are trivial and will be filtered out by the pre-processing step, it achieves worse results than other split ranges. We also provide the additional result of the DIPS-Plus test set in the appendix.

### 4.5 VISUALIZATION

We also visualize some prediction results of GeoTrans in Figure 3. We exhibit a homologous multimer (i.e., PDB ID: 4LIW) and a heterologous multimer (i.e., PDB ID: 4DR5) from the DIPS-Plus

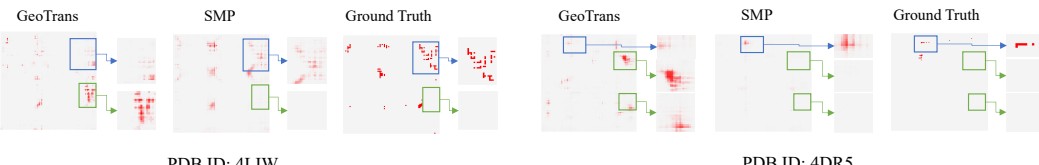

| GeoTrans | SMP | Ground Truth | GeoTrans | SMP | Ground Truth |

PDB ID: 4LIW        PDB ID: 4DR5

Figure 3: Contact visualization results of the 2 multimer 4LIW (left) and 4DR5 (right). The Geo-Trans's predictions, SMP's predictions, and ground truths correspond to the left, middle, and right columns, respectively (best viewed in color).

Table 7: Protein interaction site prediction results on four datasets (%).

| Method | Test_60 (Yuan et al., 2022) | | | | Test_315 (Yuan et al., 2022) | | | |
|---|---|---|---|---|---|---|---|---|
| | F1 | MCC | AUROC | AUPRC | F1 | MCC | AUROC | AUPRC |
| PSIVER (Murakami & Mizuguchi, 2010) | 0.0 | 0.0 | 57.83 | 18.92 | 0.0 | 0.0 | 55.96 | 16.52 |
| DeepPPISP (Zeng et al., 2020) | 3.21 | 5.08 | 64.05 | 23.75 | 4.87 | 8.13 | 66.96 | 25.41 |
| GraphPPIS (Yuan et al., 2022) | 22.90 | 24.00 | 77.89 | 41.89 | 30.26 | 27.54 | 79.62 | 40.70 |
| GraphPPIS + SMP | 28.76 | 27.82 | 78.04 | 42.39 | 30.71 | 27.93 | 79.45 | 41.02 |
| GraphBind (Xia et al., 2021) | 38.98 | 31.86 | 77.38 | 40.88 | 43.31 | 33.64 | 78.77 | 40.93 |
| GraphBind + SMP | 48.76 | 43.90 | 86.47 | 57.43 | 53.50 | 47.28 | 87.55 | 58.13 |
| | Btest_31 (Yuan et al., 2022) | | | | UBtest_31 (Yuan et al., 2022) | | | |
| PSIVER (Murakami & Mizuguchi, 2010) | 0.0 | 0.0 | 59.92 | 16.53 | 0.0 | 0.0 | 59.35 | 15.82 |
| DeepPPISP (Zeng et al., 2020) | 3.45 | 5.62 | 65.75 | 21.66 | 3.02 | 5.06 | 66.07 | 20.94 |
| GraphPPIS (Yuan et al., 2022) | 21.13 | 22.02 | 79.06 | 38.16 | 17.13 | 15.39 | 76.18 | 30.58 |
| GraphPPIS + SMP | 26.07 | 26.44 | 79.14 | 38.70 | 23.01 | 20.74 | 76.37 | 31.61 |
| GraphBind (Xia et al., 2021) | 37.76 | 33.12 | 77.58 | 40.68 | 31.60 | 26.48 | 76.42 | 35.54 |
| GraphBind + SMP | 43.95 | 40.36 | 85.96 | 51.60 | 30.48 | 25.84 | 80.68 | 36.15 |

Table 8: Complex prediction results on DIPS and DB5.5 datasets. ∗: Results reported in the original paper. †: Results reproduced by ourselves. Note: The first four lines are the traditional methods.

| Method | DIPS Test set (Townshend et al., 2019) | | | | DB5.5 Test set (Guest et al., 2021) | | | |
|---|---|---|---|---|---|---|---|---|
| | Complex RMSD ↓ | | Interface RMSD ↓ | | Complex RMSD ↓ | | Interface RMSD ↓ | |
| | Mean | Std | Mean | Std | Mean | Std | Mean | Std |
| ATTRACT (de Vries et al., 2015) | 14.93 | 10.39 | 14.02 | 11.81 | 10.09 | 9.88 | 10.69 | 10.90 |
| HDOCK (Yan et al., 2017) | 10.77 | 11.39 | 8.88 | 10.95 | 5.34 | 12.04 | 4.76 | 10.83 |
| CLUSPRO (Kozakov et al., 2017) | 14.47 | 10.24 | 13.62 | 11.11 | 8.25 | 7.92 | 8.71 | 9.89 |
| PATCHDOCK (Schneidman-Duhovny et al., 2005) | 13.58 | 10.30 | 12.15 | 10.50 | 18.00 | 10.12 | 18.75 | 10.06 |
| EQUIDOCK∗ (Ganea et al., 2021) | 14.52 | 7.13 | 11.92 | 7.01 | 14.72 | 5.31 | 13.23 | 4.93 |
| EQUIDOCK† (Ganea et al., 2021) | 15.19 | 7.71 | 12.59 | 6.31 | 15.02 | 6.10 | 14.03 | 5.60 |
| EQUIDOCK + SMP | 14.55 | 7.16 | 11.31 | 5.72 | 15.84 | 5.61 | 13.61 | 4.47 |

test set. The blue box in Figure 3 indicates that SMP can successfully predict several positive contacts that GeoTrans neglects. The green box in Figure 3 shows that our SMP can eliminate some false positives provided by GeoTrans. All these bounded areas demonstrate that SMP is more accurate in multimer contact prediction than the SOTA method GeoTrans, demonstrating that the model pre-trained by SMP can carry several types of new advantages over the original one.

### 4.6 APPLICATIONS FOR OTHER TASKS

To verify our SMP is a general method, we apply the SMP in protein interaction site prediction (PISP) and protein docking tasks, respectively. These two tasks used the same pseudo multimer data as in contact prediction. We integrate SMP with two methods GraphBind (Xia et al., 2021) and GraphPPIS (Yuan et al., 2022) to demonstrate that SMP could be effectively applied to the PISP task. It shows that SMP achieves better results than the past methods in Table 7, indicating that the SMP pre-trained on the pseudo multimer data could improve protein representation to help the downstream task. For the protein docking task, we use EQUIDOCK (Ganea et al., 2021) as the baseline model and combine SMP to show the advantage in structure-related fields. From Table 8, we could observe that SMP achieves better performance on the DIPS dataset (Townshend et al., 2019) and holds comparable results on the DB5.5 dataset (Guest et al., 2021). It demonstrates that SMP could help the model obtain a more precise multimer structure with no model framework change. More implementation details and the visualization results can be seen in the appendix.

### 5 CONCLUSION

This paper introduces the Split and Merge Proxy (SMP), a simple yet effective pre-training framework for protein inter-chain contact prediction to solve the limited number of multimers by using rich monomer information. SMP splits monomer data into pseudo multimers and trains the model to merge them back together by predicting its pseudo contact interaction, which reduces the task gap between this proxy task and the final target, leading to significant performance gain. It demonstrates that splitting monomers benefits multimer contact prediction tasks. SMP could also apply to other multimer-related tasks (e.g. protein docking and protein interaction site), which achieves better results than the previous methods. It shows that SMP is a general method that has the potential for other downstream computational multimer tasks.

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
