# APPENDIX FOR

# SPLIT AND MERGE PROXY: PRE-TRAINING PROTEIN INTER-CHAIN CONTACT PREDICTION BY MINING RICH INFORMATION FROM MONOMER DATA

## A  COMPARISON WITH THE ALPHAFOLD SERIES MODEL

In order to demonstrate the effectiveness of SMP, we compare our SMP with the AlphaFold series model in the contact prediction task. Since AlphaFold series have a head that predicts residue-residue distance which can be transferred to contact prediction, here, we conduct several experiments to compare our method with AlphaFold-Multimer (Evans et al., 2021) (AF-Multimer) and AlphaFold2 (Jumper et al., 2021) (AF2). Mention that although AF2 is designed for monomer tasks, it still has the potential to treat multimer tasks by directly concatenating multimer sequences as the input monomer shown in the Bryant et al. (2022); Gao et al. (2022), so we also decide to compare with it. We design our experiments on the DIPS-Plus, CASP-CAPRI, and DB5 benchmarks with results shown in Table 1, 2, and 3. The experiment shows that our SMP achieves better results than both AF2 and AF-Multimer, indicating the effectiveness of our SMP. Please note that we do not compare AF-Multimer on DIPS-Plus and DB5 datasets because the training set of such dataset has been included in the training data of AF-Multimer.

Table 1: Performance of AF2 and SMP on DIPS-Plus test dataset.

| | 16 (Homo) | | | 16 (Hetero) | | |
|---|---|---|---|---|---|---|
| Method | P@ $L/10$ | P@ $L/5$ | P@ $L/2$ | P@ $L/10$ | P@ $L/5$ | P@ $L/2$ |
| AF2 (Jumper et al., 2021) | 12.78 | 10.67 | 8.72 | 0.27 | 0.62 | 0.43 |
| SMP | **39.81** | **33.33** | **26.02** | **20.99** | **20.07** | **15.00** |
| | 32 (All Proteins) | | | | | |
| Method | P@ $L/10$ | P@ $L/5$ | P@ $L/2$ | R@ $L$ | R@ $L/2$ | R@ $L/5$ |
| AF2 (Jumper et al., 2021) | 6.53 | 5.64 | 4.57 | 1.92 | 1.40 | 0.63 |
| SMP | **30.40** | **26.70** | **20.51** | **24.00** | **16.02** | **8.56** |

Table 2: Performance of AF2, AF-Multimer, and SMP on CASP-CAPRI 13 & 14 dataset.

| | 14 (Homo) | | | 5 (Hetero) | | |
|---|---|---|---|---|---|---|
| Method | P@ $L/10$ | P@ $L/5$ | P@ $L/2$ | P@ $L/10$ | P@ $L/5$ | P@ $L/2$ |
| AF2 (Jumper et al., 2021) | 6.34 | 3.70 | 2.32 | 0.0 | 0.0 | 0.77 |
| AF-Multimer (Evans et al., 2021) | 14.06 | 7.43 | 3.86 | 0.0 | 0.0 | 0.0 |
| SMP | **18.63** | **14.37** | **11.57** | **32.00** | 23.49 | **18.35** |
| | 19 (All Proteins) | | | | | |
| Method | P@ $L/10$ | P@ $L/5$ | P@ $L/2$ | R@ $L$ | R@ $L/2$ | R@ $L/5$ |
| AF2 (Jumper et al., 2021) | 4.67 | 2.73 | 1.91 | 1.24 | 1.18 | 0.99 |
| AF-Multimer (Evans et al., 2021) | 10.36 | 5.47 | 2.85 | 2.47 | 2.31 | 2.19 |
| SMP | **21.97** | **16.77** | **13.36** | **14.33** | **8.34** | **3.91** |

Table 3: Performance of AF2 and SMP on DB5 test dataset.

| | 55 (Hetero) | | | | | |
|---|---|---|---|---|---|---|
| Method | P@ $L/10$ | P@ $L/5$ | P@ $L/2$ | R@ $L$ | R@ $L/2$ | R@ $L/5$ |
| AF2 (Jumper et al., 2021) | 0.074 | 0.074 | 0.054 | 0.17 | 0.063 | 0.034 |
| SMP | **1.78** | **1.88** | **1.55** | **2.53** | **1.45** | **0.69** |

## B  DISCUSSION ABOUT POTENTIAL DATA LEAKAGE

To verify whether there is a data leakage between the pseudo multimer data and real multimer data, we remove the similar sequence between the pseudo data and the DIPS-Plus test set because the similar sequence may have a similar structure. In particular, we use BLAST (Altschul et al., 1990)

to calculate the sequence identity value between the pseudo data and the DIPS-Plus test set. The E-value is set as 1e-5 in BLAST following the previous methods (Zhang et al., 2022; Kerfeld & Scott, 2011) to filter out some unreliable calculations. The average sequence identity value of each sample in the DIPS-Plus test set is listed in Table 4. The other data (14 samples) in the DIPS-Plus test set have no identity value, so we do not list them. It shows that some chains in pseudo data have high sequence identity with the DIPS-Plus test set, which could also prove that the monomer data benefits the multimer tasks and can provide an additional way to achieve the multimer data. Therefore, we remove these pseudo data with more than 30 % sequence identity value with DIPS-Plus test set by using MMseqs2 (Steinegger & Söding, 2017) and then utilize the new filtered pseudo data to re-pretrain the contact model. The results are shown in Table 5. It shows that the filtered pseudo data could achieve comparable performance as the original pseudo data and still has better results than the state-of-the-art method GeoTrans.

Table 4: The average sequence identity value of each sample in the DIPS-Plus test set.

| PDB ID | Num of searched pseudo sample | Avg seq identity | PDB ID | Num of searched pseudo sample | Avg seq identity |
|---|---|---|---|---|---|
| 4heq | 54 | 43.31 % | 1uwa | 1 | 26.00 % |
| 1uzn | 36 | 31.47% | 4dr5 | 2 | 67.00% |
| 4liw | 2 | 54.00% | 1be3 | 4 | 57.00% |
| 3re3 | 4 | 46.50% | 3a6n | 2 | 56.50% |
| 3bxs | 21 | 84.24% | 3tuy | 66 | 30.26% |
| 2g3o | 25 | 27.56% | 3t1y | 1 | 55.00% |
| 1sdu | 21 | 84.19% | 3mnn | 2 | 77.5% |
| 4to9 | 1 | 68.00% | 3jrm | 1 | 26.00% |
| 1bhn | 10 | 60.00% | 1aon | 1 | 38.00% |

Table 5: The results of filtered pseudo multimer data on DIPS-Plus test set.

| | 32 (All proteins) | | | | | | | |
|---|---|---|---|---|---|---|---|---|
| Method | P@ $L/10$ | P@ $L/5$ | P@ $L/2$ | P@ $L$ | R@ $L$ | R@ $L/2$ | R@ $L/5$ | R@ $L/10$ |
| GeoTrans (Morehead et al., 2022) | 19.00 | 17.00 | - | - | 15.00 | 9.00 | 4.00 | - |
| SMP (ori) | 30.40 | 26.70 | 20.51 | 15.87 | 24.00 | 16.02 | 8.56 | 4.79 |
| SMP (filter) | 29.13 | 25.83 | 19.26 | 14.94 | 20.70 | 13.79 | 7.08 | 4.12 |

## C  FURTHER IMPLEMENTATION DETAILS IN TRAINING

We implement the SMP using the same network architecture and hyper-parameters as Geo-Trans (Morehead et al., 2022). Specifically, we used a 2-layer graph transformer with batch normalization for the graph encoder and a 14-layer dilated residual network for the interleave module, the detailed hyper-parameters are shown in Table 6.

Table 6: The detailed hyper-parameters of SMP for the contact prediction task.

| Hyper-parameter | Value |
|---|---|
| Number of graph transformer layers | 2 |
| Hidden dimension of graph transformer | 128 |
| Number of attention head of graph transformer | 4 |
| Number of dilated residual layers | 14 |
| Kernel size of dilated residual network | $3 \times 3$ |
| Learning rate | $1e^{-3}$ (DIPS-Plus & CASP-CAPRI), $1e^{-5}$ (DB5) |
| Weight decay | $1e^{-2}$ |
| Pre-training batch size | 48 |
| Fine-tuning batch size | 48 (DIPS-Plus & DB5), 32 (CASP-CAPRI) |
| Dropout ratio | 0.2 |
| Number of early-stopping epoch | 5 |
| Number of max epoch | 50 |

## D  IMPLEMENTATION DETAILS FOR OTHER TASKS

In the original paper, we expand our SMP into other two tasks, namely protein docking, and protein interaction site prediction. Here we give the implementation details about such two tasks.

### D.1  PROTEIN DOCKING TASK

The protein docking task aims to compute the bounded multimer structure when given a pair of unbounded multimer chains. So our key is to transfer the monomer data into a pair of unbounded

pseudo multimer chains. For a monomer, we directly treat its coordinates after split as the ground-truth label and perform a random translation and rotation of one of the split monomer chains to simulate the unbound state following EQUIDOCK (Ganea et al., 2021) as input. We also filter the protein in which the distance between each residue is more than 30 Å following the past method (Ganea et al., 2021), then we could obtain $22,557$ paired pseudo data. We utilize the EQUIDOCK (Ganea et al., 2021) as our baseline, so the hyper-parameters and network architecture are kept the same as the original EQUIDOCK (Ganea et al., 2021), which are listed in Table 7. We first pre-train the model on our pseudo multimer docking data, and then fine-tune the model on real docking data. For the DB5.5 (Guest et al., 2021), we train the model on $39,937$ training complexes of the DIPS dataset and then fine-tune it on 203 training samples.

Table 7: The detailed hyper-parameters of SMP on DIPS and DB5.5 dataset.

| hyper-parameter | Value | |
|---|---|---|
| | DIPS | DB5.5 |
| Dimension of residues embedding (node embedding) | 64 | 64 |
| Dimension of edge embedding | 27 | 27 |
| Number of IGEMN (Ganea et al., 2021) layers | 8 | 5 |
| Number of attention head of IGEMN | 50 | 50 |
| Hidden dimension of IGEMN layer | 64 | 32 |
| IEGMN layers share the parameters | False | True |
| Learning rate | $3e^{-4}$ | $1e^{-4}$ |
| Weight decay | $1e^{-4}$ | $1e^{-3}$ |
| Batch size | 10 | 10 |
| Number of early-stopping epoch | 30 | 100 |
| Number of warmup epoch | 1 | 1 |
| Number of max epoch | 10000 | 10000 |
| Dropout ratio | 0.0 | 0.0 |

## D.2 PROTEIN INTERACTION SITE PREDICTION TASK

The interaction site prediction task predicts the contact state of each residue within a single chain of the multimer. So we aim to construct the pre-training dataset of pseudo multimers with the formulation of each residue to be an interactive state if the Euclidean distance is less than 6 Å between this residue and other residues on split monomer. Following the above step, we could get $45,178$ pseudo data to pre-train the interaction site prediction task. We first pre-train the model on the pseudo interaction site data, and then fine-tune it on the real multimer data. We use GraphPPIS (Yuan et al., 2022) and GraphBind (Xia et al., 2021) as our baseline, so our SMP holds the same hyper-parameters and network architectures as these baseline models, which are shown in Table 8. We train all models on Train_335 (Yuan et al., 2022) and evaluate them on Test_60 (Yuan et al., 2022), Test_315 (Yuan et al., 2022), UBtest_31 (Yuan et al., 2022), and Btest_31 (Yuan et al., 2022) dataset to ensure a fair comparison. Note that we use a fixed threshold of $0.5$ following the past method (Yang et al., 2023) to determine whether each residue interacts or not. We do not use a dynamic threshold which is decided by maximizing the F1-score or MCC metric on the test set, because the whole test set is difficult to obtain under a real-world situation.

Table 8: The detailed hyper-parameters of SMP for interaction site prediction training.

| SMP with GraphBind training | | SMP with GraphPPIS training | |
|---|---|---|---|
| hyper-parameter | Value | hyper-parameter | Value |
| Number of early-stopping epoch | 10 | Dimension of residue embedding (node embedding) | 54 |
| Number of GNN (Wu et al., 2020) layers | 4 | Number of GCN (Chen et al., 2020) layers | 8 |
| Hidden dimension of GNN layer | 64 | Cutoff threshold distance map in pre-training | 6 Å |
| Dropout ratio | 0.5 | Cutoff threshold distance map in fine-tuning | 14 Å |
| Batch size in pre-training | 256 | Hidden dimension of GCN layer | 256 |
| Batch size in fine-tuning | 64 | Batch size | 1 |
| Learning rate in fine-tuning | $5e^{-5}$ | Learning rate | $1e^{-3}$ |
| Number of max epoch | 30 | Number of max epoch | 50 |
| Learning rate in pre-training | $2e^{-4}$ | Dropout ratio | 0.1 |
| Threshold for determining interaction state | 0.5 | Threshold for determining interaction state | 0.5 |

# E DETAILED RESULTS

## E.1 FOR PARTIAL PRE-TRAINING AND FINE-TUNING EXPERIMENTS

We have shown the experiments of partial pre-training and fine-tuning in the original papers. Here, more metrics are included and detailed results are shown in Table 9 and 10.

Table 9: Detailed partial pre-training results on DIPS-Plus test dataset.

| Row | Ratio | P@ $L/10$ | P@ $L/5$ | P@ $L/2$ | P@ $L$ | R@ $L$ | R@ $L/2$ | R@ $L/5$ | R@ $L/10$ |
|-----|-------|-----------|----------|----------|--------|--------|----------|----------|-----------|
| 1 | 0 | 19.00 | 17.00 | - | - | 15.00 | 9.00 | 4.00 | - |
| 2 | 1/5 | 18.22 | 15.58 | 13.35 | 11.06 | 16.80 | 10.48 | 4.84 | 2.76 |
| 3 | 1/4 | 18.61 | 19.02 | 15.10 | 11.76 | 17.08 | 11.14 | 5.68 | 2.90 |
| 4 | 1/3 | 24.64 | 21.36 | 16.59 | 11.92 | 16.93 | 12.08 | 6.14 | 3.35 |
| 5 | 1/2 | 26.20 | 21.29 | 15.85 | 12.77 | 18.09 | 11.50 | 6.40 | 3.84 |
| 6 | 1 | **30.40** | **26.70** | **20.51** | **15.87** | **24.00** | **16.02** | **8.56** | **4.79** |

Table 10: Detailed partial fine-tuning results on DIPS-Plus test dataset.

| Row | Ratio | P@ $L/10$ | P@ $L/5$ | P@ $L/2$ | P@ $L$ | R@ $L$ | R@ $L/2$ | R@ $L/5$ | R@ $L/10$ |
|-----|-------|-----------|----------|----------|--------|--------|----------|----------|-----------|
| 1 | 0 | 5.98 | 3.70 | 2.08 | 1.98 | 1.78 | 0.82 | 0.62 | 0.56 |
| 2 | 1/5 | 15.42 | 15.40 | 12.25 | 9.63 | 14.02 | 8.59 | 4.20 | 1.96 |
| 3 | 1/4 | 19.83 | 16.12 | 12.53 | 10.80 | 15.87 | 9.71 | 4.91 | 3.10 |
| 4 | 1/3 | 19.84 | 17.11 | 12.96 | 10.62 | 14.29 | 8.70 | 4.56 | 2.50 |
| 5 | 1/2 | 23.99 | 19.55 | 15.49 | 11.92 | 16.97 | 11.11 | 5.53 | 3.22 |
| 6 | 1 | **30.40** | **26.70** | **20.51** | **15.87** | **24.00** | **16.02** | **8.56** | **4.79** |

## E.2 FOR DIFFERENT SPLIT RANGE EXPERIMENT

We also provide additional results on the DIPS-Plus test set for the experiment of the different split ranges. The results are shown in Table 11. Such results at the test set still show that range $1/3 \sim 2/3$ has a better performance than other ranges on the test set. Because the test set of DIPS-Plus only has 32 samples, its results have a certain degree of volatility.

Table 11: Different split ranges results on DIPS-Plus test dataset.

| Row | Range | P@ $L/10$ | P@ $L/5$ | P@ $L/2$ | P@ $L$ | R@ $L$ | R@ $L/2$ | R@ $L/5$ | R@ $L/10$ |
|-----|-------|-----------|----------|----------|--------|--------|----------|----------|-----------|
| 1 | $2/5 \sim 3/5$ | 21.94 | 19.47 | 14.37 | 11.66 | 17.63 | 11.43 | 6.39 | 3.65 |
| 2 | $1/3 \sim 2/3$ | **30.40** | **26.70** | **20.51** | **15.87** | **24.00** | **16.02** | **8.56** | **4.79** |
| 3 | $1/4 \sim 3/4$ | 23.11 | 20.00 | 17.22 | 14.35 | 20.80 | 12.80 | 5.85 | 3.35 |
| 4 | $1/5 \sim 4/5$ | 23.90 | 20.75 | 14.17 | 11.36 | 16.84 | 10.71 | 6.16 | 3.40 |

# F VISUALIZATION

## F.1 PROTEIN INTERACTION SITE PREDICTION

Some visualization results are shown in Figure 1. We sample $4$ instances from Test_60 (Yuan et al., 2022), their PDB IDs are 2V9T, 4H3K, 3UVJ, and 3CQC. The yellow color in Figure 1 means the prediction results of GraphBind, the blue color in Figure 1 indicates the SMP predicts where there is interaction, and the green color in Figure 1 shows the ground truth. We could find that SMP gets more true interaction sites than GraphBind (Xia et al., 2021). It illustrates that SMP improves the protein representation ability from the pseudo multimer data, which could benefit the protein interaction site task.

## F.2 PROTEIN DOCKING

We also visualize some EQUIDOCK and SMP prediction results in Figure 2 and 3 from DIPS (Townshend et al., 2019) and DB5.5 (Guest et al., 2021) datasets. We could observe that SMP has a lower CRMSD than EQUIDOCK (Ganea et al., 2021), which demonstrates SMP could achieve a more precise multimer structure than the past methods.

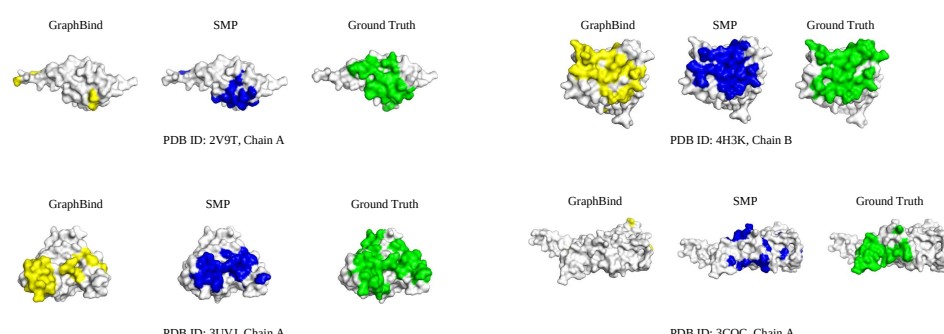

Figure 1: Interaction site visualization results on Test_60. The GraphBind's predictions, SMP's predictions, and ground truths correspond to the left, middle, and right columns, respectively.

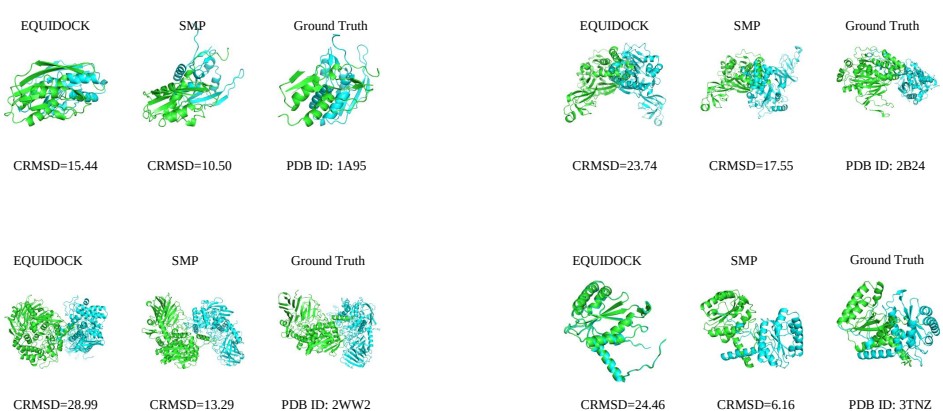

Figure 2: Docking visualization results on DIPS test set. The EQUIDOCK's predictions, SMP's predictions, and ground truths correspond to the left, middle, and right columns, respectively.

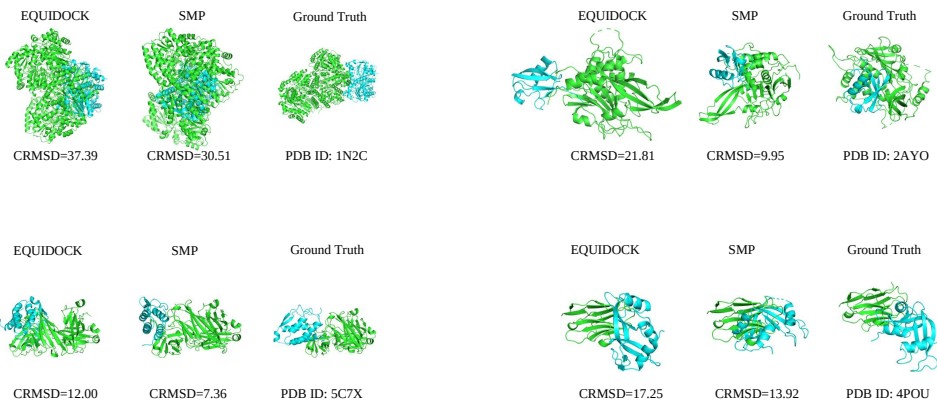

Figure 3: Docking visualization results on DB5.5 test set. The EQUIDOCK's predictions, SMP's predictions, and ground truths correspond to the left, middle, and right columns, respectively.