# OpenReview forum: "Split and Merge Proxy: pre-training protein inter-chain contact prediction by mining rich information from monomer data"
_ICLR.cc/2024/Conference — Submitted to ICLR 2024_

### Official Review · Reviewer_EkLP · 2023-10-26

**Soundness:** 3 good
**Presentation:** 3 good
**Contribution:** 3 good
**Rating:** 6
**Confidence:** 3

**Summary:**

In the context of the limited available multimer data, the authors proposed a simple yet effective method to create pseudo multimers based on monomers. To be specific, they split a monomer into two parts at a random position. They further pre-trained their model on the pseudo data with a contact prediction objective. The effectiveness of the pre-training is demonstrated through various downstream tasks.

**Strengths:**

1. The idea is novel, simple yet effective.
2. The benchmark is comprehensive.

**Weaknesses:**

1. The authors did not discuss the biological insight of their proposed pre-training method. There is no good biological sense that the interaction between two parts of a monomer could be useful to reveal the true nature of the interaction among chains of multimers.
2. Many multimers contain multiple identical chains. It might be difficult for the proposed method to be applied to modeling the interaction of those identical chains, since the pre-training objective may not be able to distinguish a chain interacting with itself and identical chains interacting with each other.

**Questions:**

1. λ the threshold for pseudo label and R the range for random split are important hyperparameters. I hope the authors could further discuss how their different values impact the pre-training process.
2. The pre-training method always splits a monomer into two parts, while it is not always the case that a multimer only consists of two chains. However, I guess the proposed method could be easily extended to more chains and I hope the authors could further discuss this issue.

---

> ### Author Response · Authors · 2023-11-23
>
> We are grateful to the reviewer for the constructive and insightful comments. Below, we provide a point-by-point reply.
>
> Q1: The authors did not discuss the biological insight of their proposed pre-training method. There is no good biological sense that the interaction between two parts of a monomer could be useful to reveal the true nature of the interaction among chains of multimers.
>
> > There are mainly two aspects of biology insight about our proposed method, and we illustrate the advantages of the interaction between two parts of a monomer (pseudo data) from the perspectives of **Richer Monomer Patterns** and **Local Part Relationships**.
> >
> > **Richer Monomer Patterns:** The pseudo data contains a wealth of monomer patterns, which can be highly beneficial for the multimer model. This can be particularly true in the case of domain-domain interactions in monomers. Studies have shown that these interactions bear similarities to the chain-chain interactions that occur in multimers [1]. Hence, the diversity and complexity of these monomer patterns can provide valuable insights for predicting multimer interactions and structures.
> > **Local Part Relationships:** Even though a random split scheme may not accurately depict the global physical relationships between two pseudo chains, it can still shed light on the local part relationships. Traditional methods for multimer contact prediction and structure prediction often compute the structure or contact based on local parts across two chains [2][3]. Therefore, despite its limitations, the random split scheme can still provide useful local patterns that can aid the computational model in predicting multimer structures.
>
>
>
> Q2: Many multimers contain multiple identical chains. It might be difficult for the proposed method to be applied to modeling the interaction of those identical chains, since the pre-training objective may not be able to distinguish a chain interacting with itself and identical chains interacting with each other.
>
> > Thank you for raising this point. On the one hand, we believe that intra-chain prediction is not the main focus of our work, and we instead aim to improve inter-chain prediction because it is a more challenging and important problem in the context of multimeric protein structure prediction. On the other hand, while we understand that the intra-chain prediction task is also important, we believe that our method can still be applied to this task. Specifically, we can use the same trained model to calculate the intra-chain interactions, and the pre-training objective can be modified to include intra-chain interactions as well. We will try this in the future.
>
> Q3: λ the threshold for pseudo label and R the range for random split are important hyperparameters. I hope the authors could further discuss how their different values impact the pre-training process.
> > Thanks for your suggestion. The range for random R can be regarded as R={$λ_1$~ $λ_2$}. So we can only focus on R and discuss different ranges on the split site as follows.
> >
> >**For R the range for random split:** Splitting at 1/5-4/5 would cause some too-short chains which are trivial and will be filtered out by the DeepInteract pre-processing step (DeepInteract requires that every single chain must have at least 20 residues). Further, splitting at other locations may lead to similar results. In the original paper, the ablation study about such splitting range shown in Table 6 is operated at the test set, which amplified the performance gap between different experiment groups because the scale of the test set (32 samples) is small. Here, we provide more stable results on the validation set (3548 samples) as the following table, shows that splitting at other locations except 1/5-4/5 would lead to similar results but 1/3-2/3 is still the best.
> >The different split range results on the DIPS-Plus validation set are shown as follows:
> >| Range   | P@L/10 | P@L/5 | P@L/2 | P@L   | R@L   | R@L/2 | R@L/5 | R@L/10 |
> >|---------|--------|-------|-------|-------|-------|-------|-------|--------|
> >| 2/5~3/5 | 46.57  | 43.62 | 36.42 | 28.16 | 41.89 | 28.91 | 14.72 | 7.84   |
> >| 1/3~2/3 | 49.82  | 46.44 | 38.02 | 28.71 | 43.28 | 30.63 | 15.84 | 8.49   |
> >| 1/4~3/4 | 49.14  | 45.66 | 37.25 | 28.25 | 42.07 | 29.58 | 15.41 | 8.27   |
> >| 1/5~4/5 | 41.90  | 39.08 | 32.34 | 24.75 | 37.53 | 25.95 | 13.30 | 7.08   |

---

> > ### Author Response · Authors · 2023-11-23
> >
> > Q4: The pre-training method always splits a monomer into two parts, while it is not always the case that a multimer only consists of two chains. However, I guess the proposed method could be easily extended to more chains and I hope the authors could further discuss this issue.
> > > Yes. Thank you for raising this point. As you said that our method is extendable to multiple chains, we believe we can split the monomer into multiple sub-chains to merge into a pseudo-multimer with multiple chains and then use them with our SMP method. However since the existing multimer benchmark for multiple chains is not well-developed and due to the limitation of computational resources, we mainly focus on the task of multimers composed of two chains in the current submission. In the future, we will further expand on multimer tasks with multiple chains.
> >
> > Reference
> >
> > [1] Sen N, Madhusudhan M S. A structural database of chain–chain and domain–domain interfaces of proteins[J]. Protein Science, 2022, 31(9): e4406.
> >
> > [2] Wang T, Yang Y, Zhou Y, et al. LRFragLib: an effective algorithm to identify fragments for de novo protein structure prediction[J]. Bioinformatics, 2017, 33(5): 677-684.
> >
> > [3] Rohl C A, Strauss C E M, Misura K M S, et al. Protein structure prediction using Rosetta[M]//Methods in enzymology. Academic Press, 2004, 383: 66-93.

---

### Official Review · Reviewer_SSrL · 2023-10-26

**Soundness:** 3 good
**Presentation:** 3 good
**Contribution:** 2 fair
**Rating:** 5
**Confidence:** 3

**Summary:**

This study introduces a novel pre-training method called "Split and Merge Proxy" (SMP) aimed at enhancing the accuracy of protein inter-chain contact prediction. Specifically, the method leverages monomer protein data to create a proxy task for model pre-training. This task involves splitting the monomer data into two sub-parts to simulate multimers and pre-training the model to merge them back together by predicting their pseudo contacts. The pre-trained model is then fine-tuned on the actual task of protein interchain contact prediction. The method demonstrates significant performance improvements over existing state-of-the-art approaches across multiple benchmark datasets, including DIPS-Plus, CASP-CAPRI, and DB5.

**Strengths:**

1. The method outperforms existing state-of-the-art methods on multiple benchmark datasets
2. The approach is simple and easy to implement and deploy.
3. The innovative agent task gives new ideas for other tasks that lack homologous pairwise data.

**Weaknesses:**

1. Data Quality: While mining information from monomer data might provide more data for multimer contact prediction, the synthesized data might not be as accurate as genuine multimeric structural data. This could lead to instability in model training or inaccuracies in prediction.

2. Challenges with Transfer Learning: While pre-training methods like SMP can leverage monomer data, the transfer from monomers to multimers might not always be seamless. This means that features in monomer data might not entirely correspond with those in multimer data.

3. Computational Complexity: Splitting and merging data could add to computational complexity, especially when dealing with large-scale protein datasets.

4. Issues with Experimental Validation: Due to the relative scarcity of genuine multimeric structural data, it might be challenging to adequately validate the model's performance. This might lead to overfitting or an over-reliance on synthesized data.

5. Structural Diversity: The diversity of protein multimeric structures might exceed what monomer data can provide, possibly limiting the model's generalization capability.

6. Other Biological Limitations: Biologically, not all monomers can simply be split and merged to simulate multimeric structures. Some proteins are highly specific in structure and function, which might impact prediction accuracy.

7. Usually, structural changes accompany the polymerization of polypeptide chains to form multimers, and the interaction between them should be distinct from the folding of single chains. In this premise, the effect remains, and the pre-training is surprisingly effective. The article lacks a discussion of this phenomenon.

**Questions:**

Would this approach also work for antibody CDR design?

---

> ### Author Response · Authors · 2023-11-23
>
> We are grateful to the reviewer for the constructive and insightful comments. Below, we provide a point-by-point reply.
>
> Q1: Data Quality: While mining information from monomer data might provide more data for multimer contact prediction, the synthesized data might not be as accurate as genuine multimeric structural data. This could lead to instability in model training or inaccuracies in prediction.
> > Although the current pseudo multimer data includes several noises, the pros outweigh the cons. Data from monomers may not be as good as true multimers, but the gain from monomer data has been demonstrated on various datasets. The main reason is that the deep learning model can neglect a certain level of data noise [1]. Since there is less real multimer data and a lot of monomer data, there is a great potential to bring better gains by improving multimer predictions with monomer data.
>
> Q2: Challenges with Transfer Learning: While pre-training methods like SMP can leverage monomer data, the transfer from monomers to multimers might not always be seamless. This means that features in monomer data might not entirely correspond with those in multimer data.
> > We agree that there is a seam between monomer and multimer. However, this seam is not quite large. Some paper has demonstrated their potential consistency [2] by directly utilizing the monomer contact prediction model in multimer scenarios. Similar results can also be found in the first row of Table 5 in our original paper, showing that the monomer model can directly show the ability to treat the multimer model. So with tranferling, our model can achieve satisfactory results in existing benchmarks.
>
>
> Q3: Computational Complexity: Splitting and merging data could add to computational complexity, especially when dealing with large-scale protein datasets.
> > The Split and merge stage is only used in training and does not affect inference. Except that, pseudo data is split offline before training, greatly reducing the computational complexity during training.
> >
> Q4: Issues with Experimental Validation: Due to the relative scarcity of genuine multimeric structural data, it might be challenging to adequately validate the model's performance. This might lead to overfitting or an over-reliance on synthesized data.
> > We think your concerns are reasonable, but experiments on a large number of multimer data show that our method works well. We previously considered whether we might be overfitting on a multimer dataset, so we evaluated it on the CASP-CAPRI and DB5 datasets in addition to the DIPS-plus dataset, which all showed the effectiveness of our method.
>
> Q5: Structural Diversity: The diversity of protein multimeric structures might exceed what monomer data can provide, possibly limiting the model's generalization capability.
> > We agree with that the monomer diversity is lower than the multimer, but we believe this will not limit the generalization of our model. The experiments in the first line of Table 5 show that only using monomer data cannot lead to a satisfied multimer model. So in our implementation, the monomer data is used for pre-training and multimer data is combined and used to fine-tine. So the information provided by the monomer is additional and does not limit the generalization capability. Actually, we have evaluated the model's generalization capability on the CASP-CAPRI and DB5. It shows our cross-set generalization capability.
> >
> Q6: Other Biological Limitations: Biologically, not all monomers can simply be split and merged to simulate multimeric structures. Some proteins are highly specific in structure and function, which might impact prediction accuracy.
> > Following biological limitations will lead to extremely small-scale data that cannot introduce satisfactory data for pretraining. Random splitting may introduce biological noise but introduce large-scale data, leading to better pre-training results. In detail, we have tried to split at the protein domain level. However, the domain-level sub-parts only occur in long protein sequences. We do statistics on ~2500 monomers in PDB dataset and find that only 5 proteins have domain sub-parts and can be split at the domain level. Such less extra data cannot benefit network pre-training. However, we find that we can simply split and merge monomers to improve multimer prediction performance.
> >

---

> > ### Author Response · Authors · 2023-11-23
> >
> > Q7: Usually, structural changes accompany the polymerization of polypeptide chains to form multimers, and the interaction between them should be distinct from the folding of single chains. In this premise, the effect remains, and the pre-training is surprisingly effective. The article lacks a discussion of this phenomenon.
> > > We think the answer is similar to your Q1 that the pros outweigh the cons and Q2 that the structure patterns between monomer and multimer have potential consistency.
> > > Except that, we think that there are other advantages introduced by our model:
> > >
> > > **Richer Monomer Patterns:** The pseudo data contains a wealth of monomer patterns which can be highly beneficial for the multimer model. This can be particularly true in the case of domain-domain interactions in monomers. Studies have shown that these interactions bear similarities to the chain-chain interactions that occur in multimers [3]. Hence, the diversity and complexity of these monomer patterns can provide valuable insights for predicting multimer interactions and structures.
> > > **Local Part Relationships:** Even though a random split scheme may not accurately depict the global physical relationships between two pseudo chains, it can still shed light on the local part relationships. Traditional methods for multimer contact prediction and structure prediction often compute the structure or contact based on local parts across two chains [4][5]. Therefore, despite its limitations, the random split scheme can still provide useful local patterns that can aid the computational model in predicting multimer structures.
> >
> >
> > Q8: Would this approach also work for antibody CDR design?
> > > Yes. Thanks for your discussion, we think that we can use our approach for antibody CDR design.  In the task of antibody CDR structure design, heavy chain and light chain interact to form a complex antibody structure, similar to the structure prediction of dimers, and the multimer structure prediction model has the potential to work in this task [6]. We can use a large number of monomers to obtain pseudo-multimers to help improve the performance of heavy chain and light chain interaction prediction. In the future, we will extend our approach to antibody CDR design.
> >
> > Reference
> >
> > [1]. Rolnick D, Veit A, Belongie S, et al. Deep learning is robust to massive label noise[J]. arXiv preprint arXiv:1705.10694, 2017.
> >
> > [2] Zeng, H., Wang, S., Zhou, T., Zhao, F., Li, X., Wu, Q. and Xu, J., 2018. ComplexContact: a web server for inter-protein contact prediction using deep learning. Nucleic acids research, 46(W1), pp.W432-W437.
> >
> >
> > [3] Sen N, Madhusudhan M S. A structural database of chain–chain and domain–domain interfaces of proteins[J]. Protein Science, 2022, 31(9): e4406.
> >
> > [4] Wang T, Yang Y, Zhou Y, et al. LRFragLib: an effective algorithm to identify fragments for de novo protein structure prediction[J]. Bioinformatics, 2017, 33(5): 677-684.
> >
> > [5] Rohl C A, Strauss C E M, Misura K M S, et al. Protein structure prediction using Rosetta[M]//Methods in enzymology. Academic Press, 2004, 383: 66-93.
> >
> > [6] Fast, accurate antibody structure prediction from deep learning on massive set of natural antibodies. Nature communications. 2023 Apr 25;14(1):2389.

---

### Official Review · Reviewer_oBJT · 2023-10-30

**Soundness:** 3 good
**Presentation:** 3 good
**Contribution:** 2 fair
**Rating:** 6
**Confidence:** 4

**Summary:**

The authors propose a novel pre-training strategy for predicting inter-chain contacts that utlizes more plentiful monomer data by splitting monomer structure and predicting the contacts in these artificially-created proxy data.  Results indicate performance improvement over state-of-the-art methods, and helps improve a method's performance compared to purely supervised training.  Multi-chain contact prediction is an extremely challenging problem, and based on the experiments provided by the authors, their method provides a good increase in performance.

**Strengths:**

The proposed pre-training method can be used to improve the performance of any method applied to multimer data. The authors demonstrate its advantage in comparison to methods trained without any pre-training, and in comparison to other structure-based pre-training methods.  The advantage over other pre-training methods is that there is no task mismatch between the pre-training and final task.

**Weaknesses:**

This is a solid submission without any real issues.

**Questions:**

Comments:

- DB5 is the most appropriate benchmark dataset, as it provides the individual chains in their unbound state.  In my opinion, using DIPS-plus data for testing is not appropriate, since the unbound structures are not available.  It is a highly valuable resource for training and validation.  Your results demonstrate how much more difficult this task is, and makes me question the value of the evaluation over the DIPS-plus and CASP-CAPRI datasets.  It also demonstrates that we are still very far from being able to accurately predict inter-chain contacts in a realistic scenario.

- The single chain proteins are split by cutting the sequence into two.  Another option would be to cluster the monomer into two clusters and use those as the proxy interacting chains.  This might give rise to more "natural" splitting of the initial monomer.  Can you comment on that?

- There is a potential concern for information leakage if the monomer data contains structures that are similar to the trained complexes.  I don't think that is a concern, since the monomer data does not contain information about the labels; infact, I would be curious to know whether allowing the model to learn from the monomers in a complex helps improve performance.

- The performance advantage for EquiDock is unclear, as the results using SMP are slightly worse than the EquiDock published results, but comparable to the result when they ran it.

Minor comments:

"However, due to the small scale of existing multimer data, current models are less accurate in protein inter-chain contact prediction."
Less accurate than existing methods?  AlphaFold-multimer will definitely be less accurate than its monomeric counterpart as complex structure prediction is a more difficult problem.

"Due to the ID numbers of monomer and multimer in PDB being different, there is no overlap between pseudo multimer and real multimer data."  One is composed of multi-chain complexes, and the other is composed of single chains, so no overlap is possible by construction.  However, the chains in the multi-chain complex data, can potentially appear in the single chain dataset.  However, I don't consider that an issue.

"we all use HHBlits (Remmert et al., 2012) with Uniclust30 (Mirdita et al., 2017) database for MSA, and PSAIA (Mihel et al., 2008) to calculate geometric features."  what do you mean by geometric features?

Please define the metric you use (e.g. P @ L/5).  Are those numbers in percentages?

"After filtering extreme data, such as too long, too short sequences and high relative data with other datasets,"  Please define "high relative data..." Did you mean sequence similarity to other datasets?  The threshold used to define that is important.  Also, I can understand why you wouldn't want to include very short sequences, but why remove long ones?
"filtering the overlap between the original CASP-CAPRI data and the DIPS-Plus." - please explain how overlap was computed.

Figure 3 is very difficult to read.

homologous multimer --> homomultimer

heterologous multimer --> heteromultimer

---

> ### Author Response · Authors · 2023-11-23
>
> Q1: DB5 is the most appropriate benchmark dataset, as it provides the individual chains in their unbound state. In my opinion, using DIPS-plus data for testing is not appropriate, since the unbound structures are not available. It is a highly valuable resource for training and validation. Your results demonstrate how much more difficult this task is, and makes me question the value of the evaluation over the DIPS-plus and CASP-CAPRI datasets.
>
> > We believe that all these benchmarks are meaningful for the inter-chain contact prediction task. Results on DIPS-plus and CASP-CAPRI show fair comparisons between our SMP and other state-of-the-art methods. Although these results cannot reflect whether the model can be applied in real scenarios or not, they still demonstrate the superiority of our model. We agree that results on DB5 show that this topic is far away from real application because of its extreme challenge. However, we still achieve a new state of the art on this dataset and plan to continue to solve that topic to make it close to real application in the future.
>
> Q2: The single chain proteins are split by cutting the sequence into two. Another option would be to cluster the monomer into two clusters and use those as the proxy interacting chains. This might give rise to more "natural" splitting of the initial monomer. Can you comment on that?
>
> > Thanks for your suggestion. We think this idea may have a main barrier in that we do not have pseudo-ground-truth contact labels for the cluster chains. We will investigate your suggestion and try to introduce it in the future.
>
> Q3: I would be curious to know whether allowing the model to learn from the monomers in a complex helps improve performance.
>
> > The monomer data would benefit the multimer models for the following reasons:
> >
> >**Richer Monomer Patterns:** The pseudo data contains a wealth of monomer patterns which can be highly beneficial for the multimer model. This can be particularly true in the case of domain-domain interactions in monomers. Studies have shown that these interactions bear similarities to the chain-chain interactions that occur in multimers [1]. Hence, the diversity and complexity of these monomer patterns can provide valuable insights for predicting multimer interactions and structures.
> >
> >**Local Part Relationships:** Even though a random split scheme may not accurately depict the global physical relationships between two pseudo chains, it can still shed light on the local part relationships. Traditional methods for multimer contact prediction and structure prediction often compute the structure or contact based on local parts across two chains [2,3]. Therefore, despite its limitations, the random split scheme can still provide useful local patterns that can aid the computational model in predicting multimer structures.
>
> Q4: The performance advantage for EquiDock is unclear, as the results using SMP are slightly worse than the EquiDock published results, but comparable to the result when they ran it.
>
> > The EQUIDOCK we reported in the original paper is our reproduced results with the EQUIDOCK official GitHub repository with the default setting. The reproduced results have a litter drop with the paper-reported one because of the device change, which is common in machine learning reproduction. We compare our SMP and EQUIDOCK under the same setting and device for fairness and to achieve better results.
> >
> > Further, to demonstrate the superiority of our SMP, we conduct the experiments in a series of less training data scenarios on the DIPS. We compare the SMP with EQUIDOCK in 40%, 30%, 20%, and 10% DIPS training data. The results are as follows:
> > ||  |         |        |     DIPS   |             |        |        |
> >|:-----------------:|:-----------------:|:-------:|:------:|:------:|:-----------:|:------:|:------:|
> >||  |         |  Complex RMSD       |        |             |    Interface RMSD    |        |
> >|Data ratio|Method |Median| Mean| Std|Median | Mean| Std|
> >|0.4|        EQUIDOCK        |  15.11    | 16.34   |  7.16  |      12.44   |13.37  |  6.56   |
> >||        SMP        |  14.78  | 15.06  | 6.72  |    11.20    | 12.59 |  6.41  |
> >|0.3|        EQUIDOCK        |    16.20   | 16.87   | 7.06   |       12.15  | 13.64 |  6.86   |
> >||        SMP        |  15.79   |  15.40 |  7.47 |   11.55     |  12.25|  6.77  |
> >|0.2|        EQUIDOCK        |    17.03   | 17.08 | 7.80  |   13.65      | 14.99 |  7.47  |
> >||        SMP        |   15.45   | 16.15  |  8.17 |   11.44    | 13.04 |  6.85  |
>  >|0.1|        EQUIDOCK        |   18.21    | 18.27 |  6.66 | 14.73        | 15.50 |  6.67  |
> >||        SMP        |   14.55   | 16.17  | 7.22 |   11.72    |  13.62|   6.57 |
> >
> > The experiments show that our SMP can adapt to different volume-level training data situations, having the potential to save the extra cost of collecting multimer data.

---

> ### Author Response · Authors · 2023-11-23
>
> Q5: "However, due to the small scale of existing multimer data, current models are less accurate in protein inter-chain contact prediction." Less accurate than existing methods?
>
> > Sorry for your confusion, we want to express that current models have low performance in the inter-chain contact prediction task because of the small-scale of existing multimer data.
>
> Q6: "we all use HHBlits (Remmert et al., 2012) with Uniclust30 (Mirdita et al., 2017) database for MSA, and PSAIA (Mihel et al., 2008) to calculate geometric features." What do you mean by geometric features?
>
> > We compute geometric features as same with the DeepInteract [4]. The geometric features include inter-residue distances, directions, and orientations with the angles between each residue pair’s amide plane normal vectors.
>
> Q7: Please define the metric you use (e.g. P @ L/5). Are those numbers in percentages?
>
> > Yes, they are in percentages. We use the same metrics as shown in the DeepInteract [4]. L means the length of shortest chain for a protein complex, `P @ L/5` represents the model’s top-L/5 precision.
>
> Q8: Figure 3 is very difficult to read.
>
> > Sorry for your confusion, we will change the size to make it clear in the final revision.
>
> Q9: homologous multimer --> homomultimer, heterologous multimer --> heteromultimer
>
> > Thanks for your suggestion. We will update these unclear descriptions in the final revision.
>
>
> Q10: "After filtering extreme data, such as too long, too short sequences and high relative data with other datasets," Please define "high relative data...". Did you mean sequence similarity to other datasets? The threshold used to define that is important
>
> > Yes, "high relative data" means that sequence similarity to other datasets. We follow DeepInteract [4] to set the threshold 30% to filter complex among our test sets and training and validation sets. The threshold of 30% sequence identity is a commonly used criterion in bioinformatics for assessing the similarity between protein sequences [5,6]. This definition will be updated in the final revision.
>
> Q11: I can understand why you wouldn't want to include very short sequences, but why remove long ones? "filtering the overlap between the original CASP-CAPRI data and the DIPS-Plus." - please explain how overlap was computed.
>
> > We follow DeepInteract [4] and filter very long sequences because they will cause severe compute consumption or even make the GPUs out of memory. We also follow DeepInteract and use the same filtered CASP-CAPRI data.
>
> Reference
>
> [1] Sen N, Madhusudhan M S. A structural database of chain–chain, and domain–domain interfaces of proteins[J]. Protein Science, 2022, 31(9): e4406.
>
> [2] Wang T, Yang Y, Zhou Y, et al. LRFragLib: an effective algorithm to identify fragments for de novo protein structure prediction[J]. Bioinformatics, 2017, 33(5): 677-684.
>
> [3] Rohl C A, Strauss C E M, Misura K M S, et al. Protein structure prediction using Rosetta[M]//Methods in enzymology. Academic Press, 2004, 383: 66-93.
>
> [4] Morehead A, Chen C, Cheng J. Geometric Transformers for Protein Interface Contact Prediction[C]//International Conference on Learning Representations. 2022.
>
> [5] Jordan, R.A., El-Manzalawy, Y., Dobbs, D. and Honavar, V., 2012. Predicting protein-protein interface residues using local surface structural similarity. BMC bioinformatics, 13(1), pp.1-14.
>
> [6] Yang, J., Roy, A. and Zhang, Y., 2013. Protein–ligand binding site recognition using complementary binding-specific substructure comparison and sequence profile alignment. Bioinformatics, 29(20), pp.2588-2595.

---

### Official Review · Reviewer_moXw · 2023-11-02

**Soundness:** 3 good
**Presentation:** 2 fair
**Contribution:** 3 good
**Rating:** 5
**Confidence:** 4

**Summary:**

This paper introduces Split and Merge Proxy (SMP), a new pretraining framework for PPI contact prediction leveraging the abundant monomer data. SMP splits monomer data into pseudo-multimers and trains the model to merge them back by predicting "inter-chain" contacts. Experiments on PPI contact prediction, protein-protein docking shows that SMP is a general method for PPI-related tasks.

**Strengths:**

1. Novelty: A novel pretraining method leveraging monomer data for multimer tasks. I believe leveraging monomer data for multimer-related tasks is important, as shown by AF2-Multimer.
2. Performance: New SOTA on benchmarks such as DIPS-Plus, CAPRI.

**Weaknesses:**

1. Writing: IMHO the writing could be significantly improved. E.g., "1.5 times more performance" should be "50% better performance", "except that" should be "in case" in your context. Also please do not shrink the margins and spacings as it makes the paper look very crowded.
2. Significance: Since AF2-Multimer and DiffDock-PP (which, by the way, should be compared in the docking benchmark) can already predict the structure of the protein complex quite well, the role of contact prediction becomes less significant. Quite similarly, in monomer structure prediction, initially contact maps are predicted and used to refine the final structure, until models like AF2 can predict protein structure end-to-end.

**Questions:**

1. When you benchmark against AF2-Multimer, are you benchmarking against the contact prediction module? Could you try computing the contact map based on the predicted structures and benchmark against that?
2. Could you compare your docking results with DiffDock-PP?
3. In response to my comment above, could you elaborate the significance of inter-chain contact prediction?

---

> ### Author Response · Authors · 2023-11-23
>
> Q1: When you benchmark against AF2-Multimer, are you benchmarking against the contact prediction module? Could you try computing the contact map based on the predicted structures and benchmark against that?
>
> > Yes, we have already employed the AlphaFold series model (including AlphaFold2 (AF2) and AlphaFold-Multimer (AF-Multimer)) for contact prediction and compared them with our SMP on the DIPS-Plus, CASP-CAPRI, and DB5 test sets in the Section A in the Supplementary Material. Here, we list the comparison results again and provide further discussions.
> >
> >|  |         |        |     DIPS-Plus   |             |        |        |
> >|:-----------------:|:-------:|:------:|:------:|:-----------:|:------:|:------:|
> >|  |         |    16(Homo)    |        |             |   16(Hetero)     |        |
> >|       Method      | P@ L/10 | P@ L/5 | P@ L/2 |     P@ L/10    | P@ L/5 | P@ L/2 |
> >|        AF2        |   12.78   |  10.67  |  8.72  |    0.27     | 0.62 |   0.43  |
> >|        SMP        |  39.81  |  33.33 |  26.02 |    20.99    | 20.07 |  15.00  |
> >|  |         |        |     32(ALL proteins)   |            |        |        |
> >|       Method      | P@ L/10 | P@ L/5 | P@ L/2 |     R@ L    | R@ L/2 | R@ L/5 |
> >|        AF2        |  6.53    |  5.64  |  4.57  |    1.92     | 1.40 |  0.63   |
> >|        SMP        |  30.40  | 26.70  |  20.51 |  24.00      | 16.02 |  8.56  |
> >
> >|  |         |        |     CASP-CAPRI   |             |        |        |
> >|:-----------------:|:-------:|:------:|:------:|:-----------:|:------:|:------:|
> >|  |         |   14(Homo)     |        |       15(Hetero)      |        |        |
> >|       Method      | P@ L/10 | P@ L/5 | P@ L/2 |     P@ L/10    | P@ L/5 | P@ L/2 |
> >|        AF2        |   6.34   |  3.70  |  2.32  |   0.0     | 0.0 |  0.77   |
> >|        AF-Multimer        |  14.06  | 7.43  | 3.86  |    0.0    |0.0  |  0.0  |
> >|        SMP        |  18.63  | 14.37  | 11.57  |  32.00     |23.49  |  18.35  |
> >|  |         |        |     19(ALL proteins)   |            |        |        |
> >|       Method      | P@ L/10 | P@ L/5 | P@ L/2 |     R@ L    | R@ L/2 | R@ L/5 |
> >|        AF2        |  4.67   | 2.73   |  1.91  |    1.24     | 1.18 | 0.99    |
> >|        AF-Multimer        |  10.36  | 5.47  | 2.85  |  2.47      | 2.31 |  2.19  |
> >|        SMP        | 21.97   | 16.77  | 13.36  |  14.33     | 8.34 | 3.91   |
> >
> >|  |         |        |     DB5   |             |        |        |
> >|:-----------------:|:-------:|:------:|:------:|:-----------:|:------:|:------:|
> >|  |         |        |    55(Hetero)    |             |        |        |
> >|       Method      | P@ L/10 | P@ L/5 | P@ L/2 |     R@ L    | R@ L/2 | R@ L/5 |
> >|        AF2        |  0.074  |  0.074  | 0.054   |    0.17    | 0.063 |  0.063   |
> >|        SMP        |  1.78  |1.88  |  1.55 |  2.53     | 1.45 | 0.69   |
> >
> > **Note:** We only compare AF-Multimer on CASP-CAPRI and neglect DIPS-Plus and DB5 datasets because the latter two ones have been included in the training data of AF-Multimer.
> >
> > **Discussions:**
> >1. The experimental results show that AF-Multimer has a better performance than AF2 in the inter-chain contact prediction task. But our SMP performs much better than both AF2 and AF-multimer.
> >2. AF2 is designed for the monomer task, so its multimer-related task cannot achieve a satisfactory result.
> >3. AF-Multimer is trained on multimer data so it can deal with multimer structure prediction. Due to the limitation of the multimer data volume, AF-Multimer still achieves worse performance than our SMP.
>
>
> Q2: Could you compare your docking results with DiffDock-PP?
>
> > We compare the DiffDock-PP [1] with our SMP on the DIPS test set for the docking task. The Vanilla DiffDock-PP is trained on a full DIPS training set (40,290 samples) for 170 epochs within 4 days by utilizing 8 A6000 GPUs (48GB). Currently, due to the limitation of rebuttal time and computation resources, we select the first 8,000 samples from the DIPS training set following a suggestion mentioned in a GitHub issue in DiffDock-PP official GitHub repository and keep other setting unchanged. The results are as follows:
> >|  |         |   DIPS     |        |
> >|:-----------------:|:-------:|:------:|:------:|
> >|  |         |  Complex RMSD       |
> >|Method |Median| Mean| Std|
> >|        DiffDock-PP        |  16.47    |  15.86  |  9.13  |
> >|        SMP        |  15.95  |  14.79 | 7.94  |
> >
> > The results show that our SMP can achieve better results than the DiffDock-PP in this scenario, which demonstrates that our SMP the has potential to help with multimer structure prediction and has the advantage in less multimer data situations. We will compare the DiffDock-PP with our SMP on the full DIPS test set in the future.
>
> Reference:
>
> [1] Ketata M A, Laue C, Mammadov R, et al. DiffDock-PP: Rigid Protein-Protein Docking with Diffusion Models[C]//ICLR 2023-Machine Learning for Drug Discovery workshop. 2023.

---

> ### Author Response · Authors · 2023-11-23
>
> Q3: Could you elaborate the significance of inter-chain contact prediction? Since AF2-Multimer and DiffDock-PP can already predict the structure of the protein complex quite well, the role of contact prediction becomes less significant. Quite similarly, in monomer structure prediction, initially contact maps are predicted and used to refine the final structure, until models like AF2 can predict protein structure end-to-end.
>
> >
> > We clarify that the importance of contact prediction should not be neglected because:
> > 1. Structure predicted by AF-Multimer is still not too accurate to meet real applications [2,3]. Inter-chain contact prediction is still useful for further accurate structure prediction, as mentioned in [4,5]. Some recent works [6,7,8] still tried to predict the multimer structure by a two-stage approach, achieving accurate results. So we believe inter-chain contact prediction does not conflict with milestones achieved by AF series.
> >
> > 2. In some scenarios, biologists may not care bout the detailed 3D structures of proteins but require the binding site and functional residues [9,10]. Thus, accurate inter-chain contact prediction can help them quickly analyze interactions between chains in a protein complex.
> >
>
>
> Q4: IMHO the writing could be significantly improved. Also please do not shrink the margins and spacings as it makes the paper look very crowded.
>
> > Thanks for your suggestion. We are working on solving all unclear descriptions and will be updated in the revision. Problems with the margins and spacings will also be treated in the revision.
>
>
> Reference
>
> [2] Zhu W, Shenoy A, Kundrotas P, et al. Evaluation of AlphaFold-Multimer prediction on multi-chain protein complexes[J]. Bioinformatics, 2023, 39(7): btad424.
>
> [3] Yin R, Feng B Y, Varshney A, et al. Benchmarking AlphaFold for protein complex modeling reveals accuracy determinants[J]. Protein Science, 2022, 31(8): e4379.
>
> [4] Lin P, Yan Y, Huang S Y. DeepHomo2. 0: improved protein–protein contact prediction of homodimers by transformer-enhanced deep learning[J]. Briefings in Bioinformatics, 2023, 24(1): bbac499.
>
> [5] Yan Y, Huang S Y. Accurate prediction of inter-protein residue–residue contacts for homo-oligomeric protein complexes[J]. Briefings in bioinformatics, 2021, 22(5): bbab038.
>
> [6] Soltanikazemi E, Roy R S, Quadir F, et al. DRLComplex: Reconstruction of protein quaternary structures using deep reinforcement learning[J]. arXiv preprint arXiv:2205.13594, 2022.
>
> [7] Quadir F, Roy R S, Soltanikazemi E, et al. DeepComplex: a web server of predicting protein complex structures by deep learning inter-chain contact prediction and distance-based modelling[J]. Frontiers in Molecular Biosciences, 2021, 8: 716973.
> [8] Baek M, Anishchenko I, Park H, et al. Protein oligomer modeling guided by predicted interchain contacts in CASP14[J]. Proteins: Structure, Function, and Bioinformatics, 2021, 89(12): 1824-1833.
>
> [9] Christopoulos A. Allosteric binding sites on cell-surface receptors: novel targets for drug discovery[J]. Nature reviews Drug discovery, 2002, 1(3): 198-210.
>
> [10] Kagaya Y, Flannery S T, Jain A, et al. ContactPFP: protein function prediction using predicted contact information[J]. Frontiers in bioinformatics, 2022, 2: 896295.

---

### Meta-Review · Area_Chair_xtvS · 2023-12-06

**Metareview:**

The paper considers the problem of predicting protein inter-chain contacts and proposes an approach to alleviate training data scarcity by splitting monomer structure and learning to merge them back, thereby enabling the use of more abundant monomer data. The proposed approach outperforms comparison methods on several datasets.

The authors provided valuable clarifying details and important additional results in their feedback to the reviewers, which would greatly improve the significance of the manuscript. In addition to incorporating these in a coherent manner, it would also be important to improve the clarity of the presentation w.r.t. the points raised by reviewer SSrL

**Justification For Why Not Higher Score:**

Although the authors provided many additional results, the paper needs significant revision is needed to incorporate the materials in a coherent and effective manner. Also important concerns raised by Reviewer SSrL deserve a deeper investigation.

**Justification For Why Not Lower Score:**

N/A

---

### Decision · Program_Chairs · 2024-01-16

Reject